



# A statistical study of the magnetic signatures of the unique Tonga volcanic explosion of 15 January 2022

Leonid F. Chernogor

[1]Department of Space Radio Physics V. N. Karazin Kharkiv National University, Kharkiv, 61022 Ukraine

*Correspondence to*: Leonid F. Chernogor (Leonid.F.Chernogor@gmail.com)

**Abstract.** For the first time, a statistical study has been conducted of the geomagnetic bay and quasi-periodic disturbances based on the datasets collected at 19 recording stations participating in INTERMAGNET Magnetic Observatories. In order to identify the disturbances from the volcanic explosion, a preliminary analysis has been used of the state of space weather during the catastrophic Tonga volcanic explosion of 15 January 2022. We summarize the main results as follows: The non-monotony of the variations in the strength of all geomagnetic field components increased appreciably on the day of the explosion as compared to the variations observed during the days used as a quiet time reference, while the eastward component of the geomagnetic field exhibited an up to 60-nT increase in variability. The duration and time delay of the bay disturbances increased with distance from the volcano, while their amplitude decreased. The propagation speeds of the bay disturbances at various observatories were determined to be in the 700–1,000 m/s range. Six groups of time delays of quasi-sinusoidal disturbances have been identified in a simultaneous analysis for the first time; they correspond to the apparent speeds of 4 km/s, 1.5 km/s, 1 km/s, as well as 500 m/s, 313 m/s, and 200 m/s. The time delay in each group increased with distance away from the volcano. The agreement between theoretical estimates and the observational data testify to the adequacy of the mechanism adopted for the generation of the disturbances.

## 1 Introduction

Five underwater Tonga volcanic explosions (20°54′ S, 175°38′ W) were observed to occur over the 04:00–05:00 UTC period on 15 January 2022, with the second explosion at 04:15 UTC being the most powerful (Adushkin et al., 2022; Astafyeva et al., 2022; Matoza et al., 2022a; Matoza et al., 2022b). The gas emissions reached 50–58-km altitude producing the highest recorded eruption column, whereas the eruption columns of Krakatoa volcano on 26–27 August 1883 reached only 40–55 km (Chernogor, 2012; McNutt et al., 2015). The Tonga volcanic eruption thermal energy is estimated to be ~$3.9 \cdot 10^{18}$ J, and the mean thermal power to be $9.1 \times 10^{12}$ W (Chernogor, 2022a; Chernogor, 2022e; Chernogor, 2023a). The mass of the erupted material attained 2.9 Gt and their volume $1.9 \times 10^9$ m³. The volcanic explosivity index (VEI) did not exceed 5.8, and the explosive energy was estimated to be in the range from 4–18 Mt of TNT to 478 ± 191 Mt of TNT (Adushkin et al., 2022; Astafyeva et al., 2022; Kulichkov et al., 2022).





The Tonga volcanic explosion was accompanied by essential disturbances in all components of the Earth
(lithosphere, ocean)–atmosphere–ionosphere–magnetosphere system (Chernogor, 2022a; Chernogor, 2022b; Chernogor,
2022c; Chernogor, 2022d; Chernogor, 2022e; Chernogor, 2023a; Chernogor, 2023b). More than 50 studies were concerned
with the effects caused by the volcanic explosion. Measurements were made of the earthquake of Richter magnitude 5.8
(Poli and Shapiro, 2022), of seismic wave propagation (Diaz et al., 2022; Matoza et al., 2022a; Matoza et al., 2022b; Poli
and Shapiro, 2022), of tsunamis (Carvajal et al., 2022; Imamura et al., 2022; Kubota et al., 2022; Ramírez-Herrera et al.,
2022; Tanioka et al., 2022; Terry et al., 2022), of Lamb waves (Kubota et al., 2022; Kulichkov et al., 2022; Lin et al., 2022;
Matoza et al., 2022a; Matoza et al., 2022b; Otsuka et al., 2022), of atmospheric gravity, infrasound, and sound waves (Burt
et al., 2022; Chen et al., 2022; Chernogor and Shevelev, 2022; Lin et al., 2022; Matoza et al., 2022a; Matoza et al., 2022b;
Wright et al., 2022), as well as observations were made of volcanic signatures in the atmosphere and ionosphere (Aa et al.,
2022a; Aa et al., 2022b; Ajith et al., 2022; Astafyeva et al., 2022; Chen et al., 2022; Chernogor et al., 2022; Harding et al.,
2022; Hong et al., 2022; Lin et al., 2022; Muafiry et al., 2022; Rakesh et al., 2022; Shinbori et al., 2022; Sun et al., 2022a;
Sun et al., 2022b; Themens et al., 2022; Zhang et al., 2022a; Zhang et al., 2022b).
Theoretical studies of the chain of physical processes were performed by (Chernogor, 2012; Chernogor, 2022a;
Chernogor, 2022b; Chernogor, 2022c; Chernogor, 2022d; Chernogor, 2022e; Chernogor, 2023a; Chernogor, 2023b).
Sun et al. (2022b) have estimated disturbances in the electric current in the ionospheric $E$ region caused by the
Tonga volcanic explosion by making use of the data on geomagnetic field variations acquired by the global network of
magnetometers. The $E$-region current density was estimated to be $J \approx 22$–55 mA/m, which changed the eastward
components, $Y$, of the geomagnetic field by ~20–50 nT. The leading front of the disturbance traveled with a propagation
speed of 740 m/s. Le et al. (2022) investigated the effect that the volcano had on the equatorial electrojet and revealed the
reversal of the electrojet direction due to a strong eastward zonal wind.
The explosion was also accompanied by variations in the geomagnetic field (Adushkin et al., 2022; Chernogor,
2023c; Chernogor and Holub, 2023a, 2023b; Iyemori et al., 2022; Le et al., 2022; Schnepf et al., 2022; Soares et al., 2022;
Yamazaki et al., 2022). Adushkin et al. (2022) have described waves and disturbances in the atmospheric electric and
magnetic fields. The data collected at 14 stations in the global network of observatories, INTERMAGNET, which are
located in the 2.790–6.225 Mm distance range from the volcano, have been used for investigating the magnetic effect. The
disturbances in the geomagnetic field have been deduced to occur on a global scale, and two groups of disturbance have been
revealed. In the first group, the disturbances were virtually synchronously observed immediately after the explosion, whereas
in the second group, the magnetic disturbances appeared after the arrival of Lamb waves. Soares et al. (2022) described
quasi-periodic disturbances in the magnitude of the eastward component, $Y$, with amplitude of ~3 nT and an ~4-min period
observed with onset time delay of 10 min at 835-km distance from the volcano. The geomagnetic variations at 3.8-mHz
(period of $T \approx 4.4$ min) have been analyzed by (Iyemori et al., 2022; Yamazaki et al., 2022), who relate these variations to
the acoustic resonance. It is important to note that the oscillations at 3.8 mHz were observed simultaneously both in the
vicinity of the volcano (API station) and in the magnetically conjugate region (HON station). The amplitudes of these





virtually synchronous oscillations were observed to be 2 nT and 0.2 nT, respectively, while the time delay of the magnetic
effect did not exceed 6 min. However, analogous oscillations were not observed at distances, $r$, greater than 2.7 Mm. The
study by Schnepf et al. (2022) is concerned with the investigation of geomagnetic variations in the 3–8-min period range
with amplitude of ~1 nT that were observed with a time delay of ~30 min (propagation speed of ~470 m/s). The authors
relate these variations to the ionospheric wave, which was generated by the volcano, and explain the variations in the 13–93-
and 5–100-min period ranges by the effects of tsunami and of atmospheric and ionospheric sources. Harding et al. (2022)
describe the multi-instrument studies of the magnetic effect of Tonga volcano. They utilized the data collected by
magnetometers at the ground and onboard the ICON and Swarm spacecraft to study the effect that the volcanic explosion
had on neutral winds and the ionospheric dynamo current system on a global scale. Despite significant progress made in
understanding the geomagnetic field disturbances related to the Tonga volcanic explosion, a further statistical and spectral
analyses of these variations is to advance understanding of this scientific issue.

74         The purpose of this paper is to present, for the first time, the inferences of the statistical and spectral analyses of the

bay and quasi-periodic disturbances in the geomagnetic field that were observed to occur after the Tonga volcanic explosion
on 15 January 2022. The data used for this research have been acquired at nineteen INTERMAGNET observatories closest
to the volcano.
**1 Information on Tonga volcano**
Tonga volcano is located ~200 m below the oceanic surface. An intense volcanic eruption was recorded to occur from
~04:00 UTC to ~16:00 UTC on 15 January 2022 when the rates of eruption attained 67 kt/s or 44,000 m$^3$/s. In total, the
volcano was active for over 12 ± 1 h, whereas the energy of the blast wave was estimated to be 16–18 Mt TNT [Chernogor,
2022a; Chernogor, 2022e; Chernogor, 2023a]. Generally, Tonga volcano is among the five most powerful on record (Table

83    1).

Table 1. Basic information on volcanos.

| Information | Krakatoa | St. Helen | El Chichón | Pinatubo | Tonga |
|---|---|---|---|---|---|
| Date | 26–27 August 1883 | 18 May 1980 | 29 March and 3–4 April 1982 | 15 June 1991 | 15 January 2022 |
| Country, location | Indonesia | USA, Skamania County | Mexico | Philippines | Kingdom of Tonga |
| Geographic coordinates | 6°06′ N, 105°25′ E | 46°12′ N, 122°11′ W | 17°22′ N, 93°14′ W | 15°7.8′ N, 120°21′ E | 20°54′ S, 175°38′ W |
| Total eruptive mass (kg) | $2.9 \times 10^{13}$ | $1.3 \times 10^{12}$ | $1.3 \times 10^{12}$ | $1.3 \times 10^{13}$ | $2.9 \times 10^{12}$ |





| Eruption column height (km) | 40–55 | 19–25 | 30–32 | 33 | 50–58 |
|---|---|---|---|---|---|
| Mean mass flow rate (kg/s) | $5.5 \times 10^8$ | $2 \times 10^7$ | $1.5 \times 10^8$ | $8 \times 10^8$ | $6.7 \times 10^7$ |
| VEI | 6 | 5 | 5 | 6 | 5–6 |
| Magnitude | 6.5 | 5.1 | 5.1 | 6.1 | 5.5 |
| Intensity | 11.7 | 10.3 | 11.2 | 12 | 10.8 |
| Notes | Current altitude 813 m, Vent 120 m | Altitude 2,549 m, Reduced by 400 m | Altitude 1,150 m, Vent 1,000 m, Crater depth 300 m | Altitude 1,486 m; before 1991, 1,745 m | Plinian underwater eruption at a 200 m depth |

The volcanic magnitude can be estimated using the following formula of McNutt et al. (2015):
$M = \log m - 7$
where $m$ is the erupted mass (in kg). Substituting $m = 2.9 \times 10^{12}$ kg yields $M \approx 5.5$, whereas the most powerful Krakatoa
volcanic has a magnitude of $M \approx 6.5$ (Table 1). The mass eruption rate, $\dot{m}$, is characterized by the intensity, given by the
relation (McNutt et al., 2015):
$I = \log \dot{m} + 3$.
Here $\dot{m}$ is in kg/s. Given the averaged value of the mass eruption rate $\dot{m} \approx 6.7 \times 10^7$ kg/s, the intensity is $I \approx 10.8$, whereas
$I \approx 11.7$ for Krakatoa volcano (Table 1).
**3 Analysis of the state of space weather**
The state of space weather over the 12–18 January 2022 period is characterized by the data retrieved from the World Data
Center for Geomagnetism, Kyoto https://wdc.kugi.kyoto-u.ac.jp/ and from the Goddard Space Flight Center Space Physics
Data Facility https://omniweb.gsfc.nasa.gov/form/dx1.html. The sunspot number did not exceed ~100, while the daily 10.7
cm solar radio flux ($F_{10.7}$) was in the ~100–120 sfu range (1 sfu = $10^{-22}$ W /(Hz m$^2$)). A substantial increase (by a factor of a
few times) in the solar wind parameters took place during the 14/15 January 2022 UTC night. The interplanetary magnetic
field $B_z$ component showed a decrease from ±4 nT to –14 nT, the equatorial $D_{st}$ index exhibited a decrease from 10 nT to –90
nT, while the auroral activity index $A_p$ showed an increase from ~5 nT to 67 nT, and the 3-h range planetary $K_p$ index from
~1 to 5.7. Thus, a *G2*-moderate *geomagnetic storm took place over the* 14/15 January 2022 night. The recovery phase of the
storm proceeded over the 15–18 January 2022 period. It should be noted that the auroral electrojet (AE) index was observed
to be ~100 nT over 04:00–11:00 UTC period on 15 January 2022, i.e., geomagnetic conditions were quiet, whereas before



04:00 UTC and after ~11:00–12:00 UTC, the AE index exceeded 500 nT, which indicated a geomagnetic disturbance
(substorm). 13 January 2022, when the geomagnetic conditions were quietest, was chosen to be a quiet time reference. 17
January 2022 is also used, although partially, as a quiet time reference.

## 4 Instrumentation and techniques

The study is based on data from the INTERMAGNET magnetic observatory network, which were accessed through the
https://www.intermagnet.org/. The list of the stations is presented in Table 2, and their locations around Tonga volcano are
depicted in Figure 1. It is important to note that the stations are located around all cardinal points as seen from the volcano.
We have analyzed the temporal variations in the northward, $X$, eastward, $Y$, and vertical, $Z$, components of the geomagnetic
field acquired on 12, 13, 15, 16, 17, and 18 January 2022 with 1-min temporal resolution and the root-mean-square error not
exceeding 1 nT.
The algorithm for finding the geomagnetic field response to Tonga volcanic explosion is as follows:
(1) Since the variations in the geomagnetic field may be caused by many powerful sources releasing significant
amounts of energy, any characteristic changes in the variations in the strength of the $X$, $Y$, and $Z$ components that were
observed to occur after the volcanic explosion and could be associated with the explosion are highlighted at the first stage of
employing the algorithm. This condition is necessary but insufficient.
(2) At the second stage, the variations analogous to those that occurred on quiet time days and were due to, for
example, diurnal variation, the solar terminator, etc., are filtered out.
(3) Next, the possible time delays and apparent speeds are determined. The time delay should increase with distance
from the volcano.
(4) If some apparent speeds at different stations are substantially close to each other, they are included in a
particular statistic. The closeness of the apparent speeds in this particular statistic is considered a sufficient condition for this
particular disturbance to be due to the volcanic explosion.
(5) The physical significance of the apparent speeds is an additional sufficient condition: these speeds must
correspond to the known speeds of waves of particular physical nature.
(6) The results obtained are compared, if possible, with the results obtained for the volcanoes that exploded before.
It should be emphasized that the variations in the geomagnetic field components were generally more or less
smooth on the days used as a quiet time reference, whereas they became non-monotonical after the Tonga volcanic explosion
when aperiodic and quasi-periodic variations were observed to occur in the magnitude of the geomagnetic field components.
The moving average process was first created by averaging, over 60-min intervals, the raw data $X(t)$, $Y(t)$, and $Z(t)$ sampled
at a 1-min time step to be subtracted from the temporal variations in the raw data to yield the $\Delta X(t)$-, $\Delta Y(t)$-, and $\Delta Z(t)$-
component deviations, which were finally subjected to the Fourier and wavelet transforms (Chernogor, 2008).



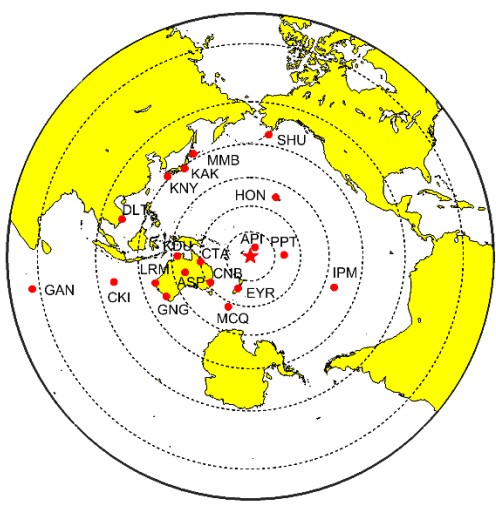


**Figure 1: Map showing the sites of the recording stations. Star designates the volcano.**

Table 2. Information on the INTERMAGNET magnetic observatories.

| IAGA[a] station code | Geographic latitude and longitude | Magnetic latitude and longitude | Country | Distance from the explosion (km) |
|---|---|---|---|---|
| Apia (API) | 13.8155° S 171.7812° W | 15.01° S 96.77° W | Western Samoa | 840 |
| Pamatai (Papeete) (PPT) | 17.5670° S 149.5740° W | 15.15° S 74.29° W | French Polynesia | 2,730 |
| Eyrewell (EYR) | 43.4740° S 172.3930° E | 46.56° S 106.28° W | New Zealand | 2,790 |
| Canberra (CNB) | 35.3200° S 149.3600° E | 41.75° S 132.81° W | Australia | 3,806 |
| Charters Towers (CTA) | 20.0900° S 146.2640° E | 27.05° S 138.47° W | Australia | 3,990 |
| Macquarie Island | 54.5000° S | 59.32° S | Australia | 4,349 |

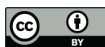



| | | | | |
|---|---|---|---|---|
| (MCQ) | 158.9500° E | 116.38° W | | |
| Honolulu (HON) | 21.3200° N | 21.65° N | United States of America | 5,024 |
| | 158.0000° W | 88.98° W | | |
| Alice Springs (ASP) | 23.7620° S | 31.83° S | Australia | 5,210 |
| | 133.8830° E | 151.19° W | | |
| Kakadu (KDU) | 12.6900° S | 20.96° S | Australia | 5,602 |
| | 132.4700° E | 153.66° W | | |
| Isla de Pascua Mataveri (IPM) | 27.1713° S | 19.48° S | Chili | 6,675 |
| | 109.4200° W | 34.44° W | | |
| Gingin (GNG) | 31.3560° S | 40.34° S | Australia | 6,887 |
| | 115.7150° E | 170.60° W | | |
| Learmonth (LRM) | 22.2200° S | 31.28° S | Australia | 7,233 |
| | 114.1000° E | 172.67° W | | |
| Kakioka (KAK) | 36.2320° N | 28.13° N | Japan | 7,852 |
| | 140.1860° E | 150.18° W | | |
| Kanoya (KNY) | 31.4200° N | 22.70° N | Japan | 8,135 |
| | 130.8800° E | 158.28° W | | |
| Memambetsu (MMB) | 43.9100° N | 36.09° N | Japan | 8,265 |
| | 144.1900° E | 147.57° W | | |
| Shumagin (SHU) | 55.3500° N | 54.46° N | United States of America | 8,557 |
| | 160.4600° W | 100.96° W | | |
| Dalat (DLT) | 11.9400° N | 2.60° N | Vietnam | 9,068 |
| | 108.4800° E | 178.89° W | | |
| Cocos (Keeling) Islands (CKI) | 12.1875° S | 21.21° S | Australia | 9,308 |
| | 96.8336° E | 168.97° E | | |
| Gan International Airport (GAN) | 0.6946° S | 8.34° S | Maldives | 12,210 |
| | 73.1537° E | 145.40° E | | |

ᵃIAGA stands for International Association of Geomagnetism and Aeronomy





## 5 Analysis of temporal variations in geomagnetic field strengths

A preliminary analysis of the temporal dependences $X(t)$, $Y(t)$, $Z(t)$ and of their time derivatives $\dot{X}(t)$, $\dot{Y}(t)$, $\dot{Z}(t)$ determined that the character of the variations on 15 January 2022 was markedly different from that observed during the quiet time reference periods when the variations were smoother and the values of the derivatives were noticeably smaller.

*API Station.* Geomagnetic bay disturbances were absent on 13 January 2022 (Figure 2), and the magnitude of fluctuations did not exceed 1 nT. On 17 January 2022, used as a quiet time reference, synchronous geomagnetic bay disturbances were absent (Figure 2). The magnitudes of fluctuations in all components exhibited insignificant variability within the $\pm 1$-nT limits.

On 15 January 2022, the geomagnetic bay disturbances appeared with a time delay, $\tau$, of ~16 min and lasted for $\Delta T_X \approx 120$ min, $\Delta T_Y \approx 146$ min, and $\Delta T_Z \approx 130$ min. They were observed to occur virtually synchronously in all three components of the geomagnetic field (Figure 2). The peak deviations from the trend in the bay disturbances are estimated to be $\Delta X \approx 16$ nT, $\Delta Y \approx 26$ nT, and $\Delta Z \approx -13$ nT. The more rapid fluctuations are superimposed upon these slow enough variations; they appear with time delays of $\Delta t_0 \approx 6$ min, $\Delta t_1 \approx 8.5$ min, $\Delta t_2 \approx 14$ min, $\Delta t_3 \approx 19$ min, $\Delta t_4 \approx 33$ min, $\Delta t_5 \approx 50$ min, and $\Delta t_6 \approx 75$ min (Table 3).

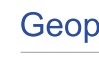

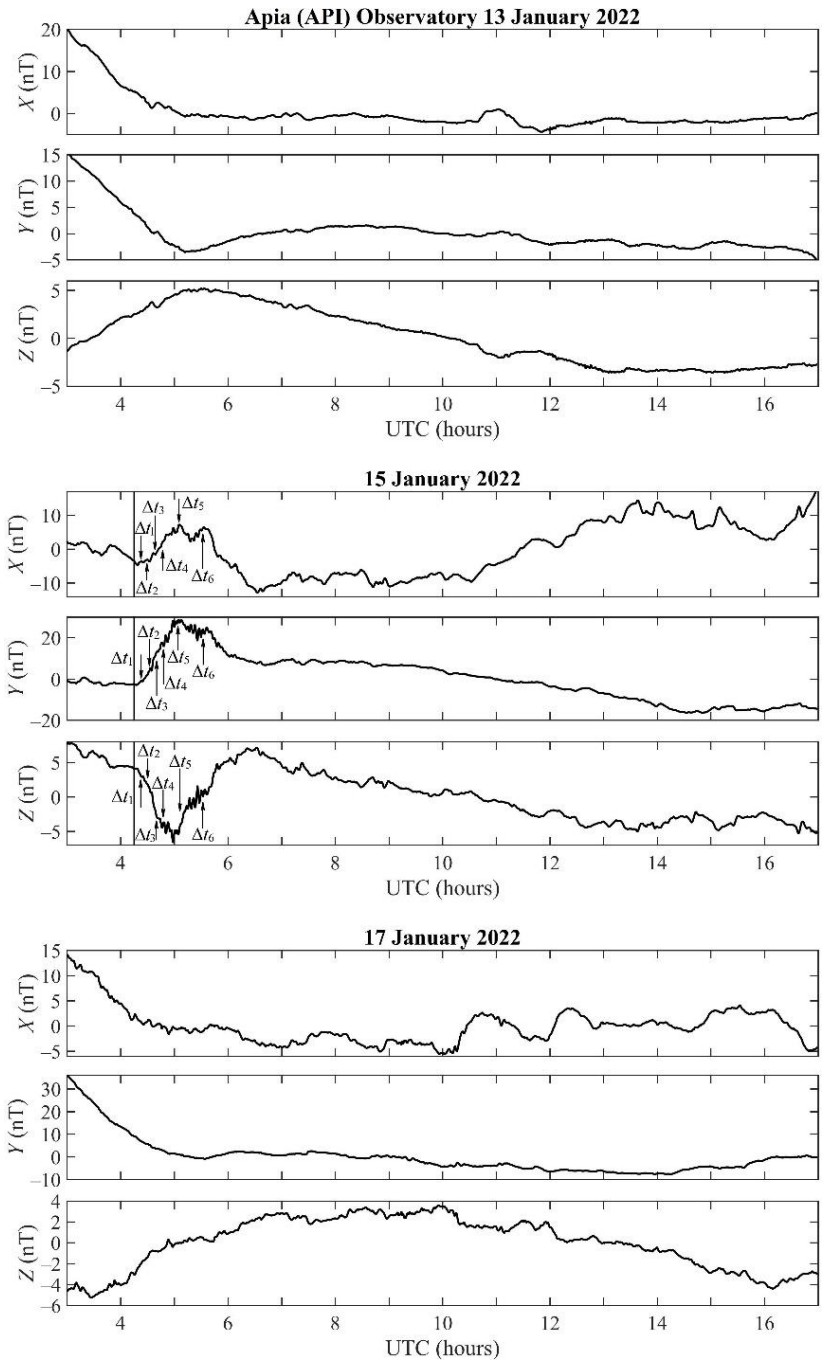

153

**Figure 2: UTC variations of the northward, *X*, eastward, *Y*, and vertical, *Z*, components of the geomagnetic field at**
**the API station during 15 January 2022, the day of the volcanic explosion, and during the days used as a quiet time**
**reference. The vertical line marks the moment of the most powerful explosion. Arrows indicate possible moments of**
**the onset of the magnetic field response.**





Table 3. Time delays and apparent speeds of disturbances in the geomagnetic field.

| Station | $\Delta t_1$ (min) | $v'_1$ (m/s) | $\Delta t_2$ (min) | $v'_2$ (m/s) | $\Delta t_3$ (min) | $v'_3$ (m/s) | $\Delta t_4$ (min) | $v'_4$ (m/s) | $\Delta t_5$ (min) | $v'_5$ (m/s) | $\Delta t_6$ (min) | $v'_6$ (m/s) |
|---|---|---|---|---|---|---|---|---|---|---|---|---|
| API | 8.5 | 4,000 | 14 | 1,560 | 19 | 1,000 | 33 | 500 | 50 | 311 | 75 | 200 |
| PPT | 16 | 4,100 | 37 | 1,420 | 50 | 1,011 | 96 | 500 | 150 | 314 | 235 | 198 |
| EYR | 17 | 3,875 | 38 | 1,410 | 50 | 1,033 | 97 | 505 | 155 | 310 | 240 | 200 |
| CNB | 21 | 4,000 | 47 | 1,500 | 68 | 1,006 | 130 | 507 | 208 | 312 | 322 | 200 |
| CTA | 22 | 3,900 | 49 | 1,510 | 71 | 1,008 | 137 | 504 | 217 | 314 | 338 | 200 |
| MCQ | 23 | 4,030 | 53 | 1,510 | 77 | 1,007 | 150 | 500 | 237 | 313 | 368 | 200 |
| HON | 26 | 4,000 | 61 | 1,490 | 89 | 1,000 | 173 | 498 | 272 | 313 | 424 | 200 |
| ASP | 27 | 3,950 | 63 | 1,500 | 92 | 1,000 | 185 | 482 | 282 | 313 | 440 | 200 |
| KDU | 28 | 4,060 | 67 | 1,500 | 98 | 1,004 | 190 | 505 | 305 | 311 | 475 | 199 |
| IPM | 33 | 3,970 | 79 | 1,500 | 115 | 1,011 | 215 | 530 | 360 | 313 | 565 | 199 |
| GNG | 34 | 4,000 | 82 | 1,500 | 119 | 1,007 | 235 | 500 | 372 | 312 | 580 | 200 |
| LRM | 35 | 4,018 | 85 | 1,500 | 125 | 1,000 | 245 | 500 | 390 | 313 | 615 | 198 |
| KAK | 38 | 3,967 | 90 | 1,540 | 135 | 1,007 | 260 | 513 | 490 | 315 | 645 | 204 |
| KNY | 39 | 3,988 | 95 | 1,507 | 140 | 1,004 | 270 | 512 | 435 | 315 | 685 | 199 |
| MMB | 39.5 | 3,993 | 97 | 1,497 | 143 | 998 | 273 | 514 | 442 | 315 | 690 | 201 |
| SHU | 40 | 4,070 | 100 | 1,501 | 147 | 1,004 | 285 | 509 | 460 | 313 | 720 | 199 |
| DLT | 43 | 3,976 | 106 | 1,501 | 156 | 1,001 | 305 | 504 | 488 | 313 | 760 | 200 |
| CKI | 44 | 3,978 | 110 | 1,477 | 160 | 1,001 | 310 | 509 | 500 | 313 | 780 | 200 |
| GAN | 56 | 3,990 | 140 | 1,507 | 208 | 1,002 | 410 | 502 | 660 | 311 | 1,020 | 200 |





*PPT Station.* On 13 and 17 January 2022, used as a quiet time reference, the strength of the *X*-component showed
variations from about (–4)–(–5) nT to 2–5 nT (Figure 3) throughout the entire 03:00–17:00 UTC period, whereas the *Y*-
component increased from (–6)–(–7) nT to 7–8 nT over the 03:00–05:00 UTC period and afterwards exhibited fluctuations
within the 2–3-nT limits, gradually decreasing from ~0 nT to (–15)–(–23) nT. On 13 January 2022, the *Z*-component
exhibited undulating oscillation during the 03:00 to ~10:00 UTC period followed by a gradual decrease from ~0 to –10 nT,
whereas on 17 January 2022, it showed a broad maximum of 8 nT near ~04:00 UTC followed by a gradual decrease to a
minimum of –5.1 nT at ~10:30 UTC and later by oscillations with an amplitude of ~1 nT around the trend changing in the –
5.1- to 0-nT range.
On the day of the volcanic explosion, the non-monotonousness in the magnitude of all components increased, the
fluctuations of the components also somewhat increased, while the trend substantially smoothed in all components. The
magnitude of the *X*-component increased from –10 nT to 20 nT, the value of the *Y*-component decreased from 15 nT to –20
nT, and of the *Z*-component decreased from 10 nT to –5 nT. In addition, six groups of disturbances appeared with time
delays of $\Delta t_1 \approx 16$ min, $\Delta t_2 \approx 37$ min, $\Delta t_3 \approx 50$ min, $\Delta t_4 \approx 96$ min, $\Delta t_5 \approx 150$ min, and $\Delta t_6 \approx 235$ min (see Figure 3). The
greatest disturbances (up to 10 nT) occurred after 14:00 UTC in the *X* component.
*EYR Station.* On 13 January 2022, the strength of the *X*-component increased, fluctuating, from –5 nT to 10 nT, and
then decreased from 10 nT to –3 nT (Figure 4). Since 06:00 UTC, the magnitude of the *X*-component fluctuated around the
2-nT value. The strength of the *Y*-component first showed a decrease from 12 nT to ~0 nT during the 03:30–05:00 UTC
period, then it fluctuated around ~0 nT, and afterwards its value was observed to decrease to –(5–10) nT. The magnitude of
the *Z*-component decreased from ~8 nT to 0 nT and fluctuated around 0 nT afterwards. On 17 January 2022, the magnitude
of the *X*-component was fluctuating around 0 nT, with excursions attaining 6–7 nT, while the magnitude of the *Y*-component
was decreasing, fluctuating, from ~20 nT to –10 nT. At the same time, the strength of the *Z*-component decreased from 7–8
nT to –10 nT, and then it increased, noticeably fluctuating, from –10 nT to 7 nT.
On the day the volcanic explosion occurred, the number of groups of disturbances was observed to attain six. The
most pronounced disturbances were negative geomagnetic bay disturbances with time delays of $\tau_X \approx 86$ min, $\tau_Y \approx 51$ min and
$\tau_Z \approx 51$ min. The drops in the *X*-, *Y*-, and *Z*-components attained –39 nT, –27 nT, and –22 nT, respectively. The drops in the
*X*-, *Y*-, and *Z*-component strengths were followed up by increases of ~38 nT, 30 nT, and 30 nT, respectively. The amplitudes
of other disturbances usually did not exceed a few nanoteslas, and they arrived with time delays of $\Delta t_1 \approx 15$ min, $\Delta t_2 \approx 38$
min, $\Delta t_3 \approx 50$ min, $\Delta t_4 \approx 97$ min, $\Delta t_5 \approx 155$ min, and $\Delta t_6 \approx 240$ min (see Table 3).






**Figure 3: Same as in Figure 2 but for the PPT station.**







**Figure 4: Same as in Figure 2 but for the EYR station.**





191    *CNB Station.* On 13 January 2022, the fluctuations in the magnitude of all components did not exceed 1–4 nT

192 (Figure 5). The strength of the *Y*-component showed a decrease from ~17 nT to –5 nT. The *Z*-component was observed to

193 drop from 4 nT to –7 nT over the 03:00–05:00 UTC period, and then its magnitude showed fluctuations around 1 nT. On 17

194 January 2022, the strength of the *X*-component gradually increased from –10 nT to 10 nT, fluctuating within the ±(5–7)-nT

195 limits, while the magnitude of the *Y*-component decreased from 27 nT to –10 nT. The strength of the *Z*-component increased

196 from –12 nT to 7 nT first, and then fluctuated within the ±(3–4)-nT limits.

197    During the course of the day the volcanic explosion occurred, the trend $\overline{X}$ first increased from –10 nT to 10 nT,

198 then it decreased from 10 nT to –10 nT, and once again increased from –10 nT to 30 nT, while the magnitude of fluctuations

199 was observed to be ±(3–5) nT. The trend $\overline{Y}$ first decreased from 16 nT to –40 nT, then it increased from –40 nT to 28 nT,

200 and once again decreased from 28 nT to –30 nT. The trend $\overline{Z}$ first decreased from –8 nT to –13 nT, then it increased from –

201 13 nT to 14 nT, and once again decreased from 14 nT to –5 nT, which was followed by an increase in $\overline{Z}$ from –5 nT to 15

202 nT. The variations with amplitude of a few nanotesla were superimposed on the slow trend in all components.

203    *CTA Station.* On the days used as a quiet time reference, the magnitudes of all components exhibited relatively

204 small fluctuations (Figure 6) except for the variations in the *X*-component on 17 January 2022 when its strength showed

205 fluctuations within the ±5-nT limits while the trend $\overline{X}$ increased from –10 nT to 5 nT. Instead, the trend $\overline{Y}$ decreased from

206 25 nT to approximately –10 nT, and the trend $\overline{Z}$ from 13 nT to –5 nT. On 13 January 2022, the trend $\overline{X}$ decreased from 12

207 nT to –5 nT over the 03:00–05:00 UTC period and then remained at this level. The trend $\overline{Y}$ decreased from 7 nT to –4 nT

208 from 03:00 UTC to 07:00UTC, then increased to 2 nT, and afterwards gradually decreased from 2 nT to –5 nT. The trend $\overline{Z}$

209 sharply decreased from 8 nT to –2 nT over the 03:00 UTC to 07:00 UTC period, and then it showed fluctuations around 0

210 nT. During 13 and 17 January 2022 used as a quiet time reference, synchronous geomagnetic bay disturbances were absent

211 (Figure 6).

212    The magnitude of fluctuations in all components considerably increased on 15 January 2022. The trend $\overline{X}$

213 increased from –15 nT to 20 nT. The *Y*-component, in addition to fluctuations, exhibited a deep drop from ~15 nT to –45 nT

214 that occurred from 05:45 UTC to 08:30 UTC (see Figure 6). The *Z*-component also showed a drop from ~0 nT to –13 nT

215 during the 05:20–07:15 UTC period, followed by a surge from –13 nT to ~23 nT. In addition, the disturbances appeared in

216 all components with time delays of 22 min, 49 min, 71 min, 137 min, 217 min, and 338 min (see Figure 6).






**Figure 5: Same as in Figure 2 but for the CNB station.**






**Figure 6: Same as in Figure 2 but for the CTA station.**



*MCQ Station.* On 13 January 2022, $X(t)$, $Y(t)$, and $Z(t)$ components showed variations in the trend not exceeding 20
nT (Figure 7), as well as fluctuations within the ±3-nT range. On 17 January 2022, the strengths of all components showed
small variations up to 10:00 UTC, whereas the variations increased to 100–200 nT after 10:00 UTC.
On the day the volcanic explosion occurred, all components exhibited relatively small variations before 11:00 UTC,
whereas they showed an increase of up to 300–400 nT after 11:00 UTC. Approximately from 06:00 UTC to 08:00–09:00
UTC, the strengths of the $X$-, $Y$-, and $Z$-component decreased by 80 nT, 40 nT, and 30 nT, respectively. Such a perturbation
pertains to a bay disturbance. In addition, except for the bay disturbance, quasi-periodic disturbances occurred with strengths
of 1–10 nT and $T \approx$ 5–10-min periods.
*HON Station.* On 13 January 2022, the $X$ component exhibited weak fluctuations within the ±1 nT limits from 00:00
UTC to 10:00 UTC (Figure 8). After 10:00 UTC, the level of variability noticeably increased. The strength of the $Y$
component displayed a rise from –4 nT to 6 nT over the 00:00–04:00 UTC period, followed by a gradual decrease to 0 nT at
12:00, and the trend continued to decrease later. Throughout the 00:00–04:00 UTC period, the magnitude of the $Z$
component showed first an increase from –3 nT to 1.3 nT, and then it decreased, fluctuating, from 1.3 nT to –2 nT. On 17
January 2022, all components showed insignificant (less than 1 nT) fluctuations from 00:00 UTC to 10:00 UTC. After 10:00
UTC, the magnitude of the fluctuations increased to ±3 nT.
On 15 January 2022, the $X$ component exhibited fluctuations within the ±(5–7) nT limits, and the fluctuations in the
strength of the $Y$ component was noticeable enough (up to ±(2–4) nT) as well. The $Z$ component also showed a significant
enhancement in fluctuations after the volcanic explosion, while all geomagnetic field components exhibited several groups
of disturbance from 04:30 UTC to 12:00 UTC.
*ASP Station.* On 13 January 2022 used as a quiet time reference, the trend $\overline{X}$ first decreased from ~15 nT to –(5–0)
nT, and then it remained at a level of 0 nT (Figure 9). The trend $\overline{Y}$ decreased from ~8 nT to –5 nT, whereas the trend $\overline{Z}$
first sharply decreased from 13 nT to –3 nT, and then it exhibited variations between –3 nT and –1 nT. The strengths of
fluctuations in all components usually did not exceed 1–2 nT. On 17 January 2022 used as a quiet time reference interval, the
trend $\overline{X}$ exhibited insignificant changes, and the strength of fluctuations did not exceed ±(3–5) nT. The trend $\overline{Y}$ decreased
from 27 nT to –21 nT, and its strength showed fluctuations attaining ±(8–10) nT. The trend $\overline{Z}$ first sharply decreased from
19 nT to –5 nT, and then $\overline{Z} \approx$ –4 nT; the magnitude of fluctuations did not exceed ±1 nT.
During the day the volcanic explosion occurred, all components of the geomagnetic field experienced geomagnetic
bay disturbances that were superimposed on fluctuations with strengths of up to 4–5 nT. The trend $\overline{X}$ exhibited a drop from
0 nT to –15 nT, whereas the trend $\overline{Y}$ showed a considerably greater drop, from 12 nT to –40 nT, of almost 4 h temporal
duration; a powerful surge from –40 nT to 20 nT of 5.5 h duration followed afterwards. The trend $\overline{Z}$ first increased from –
10 nT to –2 nT, then decreased from –2 nT to –10 nT, and increased from –10 nT to 22 nT afterwards; this surge in the trend
$\overline{Z}$ was followed by a decrease to 0–5 nT.






**Figure 7: Same as in Figure 2 but for the MCQ station.**



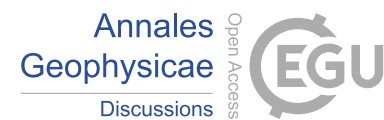




**Figure 8: Same as in Figure 2 but for the HON station.**






**Figure 9: Same as in Figure 2 but for the ASP station.**



*KDU Station*. On 13 January 2022, the trend $\overline{X}$ first sharply decreased from ~27 nT to –5 nT, and then it showed
fluctuations around a strength of –5 nT (Figure 10). At the same time, the trend $\overline{Y}$ gradually decreased from 2–3 nT to –4
nT, while the trend $\overline{Z}$ first (up to 06:30 UTC) increased to 11 nT, then sharply decreased to –5 nT and did not change
afterwards. During 17 January 2022, the behavior of the trends $\overline{X}$ and $\overline{Y}$ was qualitatively analogous to their behavior
observed on 13 January 2022; however, fluctuations in the strength increased to 2–4 nT. The trend $\overline{Z}$, fluctuating, decreased
from approximately 14 nT to –5 nT.
On the day the volcanic explosion occurred, the trend $\overline{X}$ first decreased, fluctuating, from –7 nT to –15 nT, next it
increased from –15 nT to 10 nT before ~14:00 UTC, and then a drop was observed to occur to –10 nT over the 14:00–15:30
UTC period. The trend $\overline{Y}$ first increased from –3 nT to 8 nT, next it decreased from 8 nT to –23 nT, then it increased from –
23 nT to 20–22 nT, and finally it gradually decreased from 20–22 nT to –18 nT. The trend $\overline{Z}$ first increased from ~3 nT to 9
nT, then it decreased from 9 nT to –8 nT, and once again increased from –8 nT to 13 nT. After this peak, the value of $\overline{Z}$ was
observed to gradually decrease from 13 nT to –5 nT. Variations with amplitudes of a few nanotesla were superimposed on
the relatively smooth changes in all components.
*IPM Station*. On 13 January 2022 used as a quiet time reference, the magnitude of all components before 11:00–
12:00 UTC varied within the 5–7-nT limits (Figure 11). Quasi-periodic variations were virtually absent. During 17 January
2022 up to 12:00 UTC, the variations in *X*-, *Y*-, and *Z*-components did not exceed 3–5 nT.
On 15 January 2022, the day the volcanic explosion occurred, insignificant bay reductions of only (4–8) nT in the
magnitudes of all components were observed to appear with time delays of 120–125 min and durations of 210–230 min,
whereas quasi-periodic disturbances were virtually absent.
*GNG Station*. On 13 January 2022, used as a quiet time reference, the trend $\overline{X}$ first sharply decreased from 20 nT
to –5 nT and then fluctuated around –5 nT (Figure 12). The trend $\overline{Y}$, fluctuating within the ±(3–4)-nT limits, gradually
decreased from 12 nT to –5 nT. Over the 03:00–09:00 UTC period, the trend $\overline{Z}$ substantially sharply decreased from 25 nT
to –8 nT, next it remained almost constant. On 17 January 2022, also used as a quiet time reference, the trend $\overline{X}$ gradually
increased from –12 nT to 10 nT, while the strength of the *X*-component showed fluctuations within the ±(4–10)-nT limits.
The trend $\overline{Y}$ first increased to 30 nT, and then gradually decreased from 30 nT to –20 nT showing fluctuations sometimes
attaining ±5–10 nT. The trend $\overline{Z}$ increased to 38 nT by 05:30 UTC, then decreased to –17 nT by 12:00 UTC, and later
almost did not change; the amplitude showed fluctuations within the ±4–5-nT limits after 12:00 UTC.
Throughout the day the volcanic explosion occurred, all components showed variations qualitatively different from
those observed over a quiet time period. Approximately since 06:00 UTC, all components reduced their strengths by 20–50
nT during 2–3 h. Next, their strengths increased by 15–40 nT over an almost 2-h interval. All components exhibited 5–9-nT
variations superimposed on the slow changes.





**Figure 10: Same as in Figure 2 but for the KDU Station.**



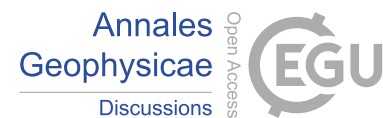




**Figure 11: Same as in Figure 2 but for the IPM Station.**







Figure 12: Same as in Figure 2 but for the GNG Station.





*LRM Station.* On 13 January 2022, the trend $\overline{X}$ nearly linearly decreased from 30 nT to –10 nT over the 03:30–
08:30 UTC period, and later it changed insignificantly, while the trend $\overline{Y}$ showed variations not exceeding ~7–8 nT (Figure
13). The trend $\overline{Z}$ first showed fluctuations about the 7-nT strength level, and later about the –5-nT level. On 17 January
2022, the trend $\overline{X}$ showed fast fluctuations within the ±10-nT limits, while the trend itself first increased to 06:00 UTC, and
then decreased from 14 nT to –10 nT before 10:00 UTC. The trend $\overline{Y}$ increased from –15 nT to 25 nT, and then decreased
nonmonotonically to –20 nT, while the amplitude of fluctuations attained ±(5–6) nT. The trend $\overline{Z}$ decreased from 40 nT to
–15 nT during the 04:00–10:00 UTC period and then fluctuated around –20 nT.
On 15 January 2022, the day the volcanic explosion occurred, the variations were observed to be substantially
different. After exhibiting insignificant fluctuations from 03:00 UTC to 06:00 UTC, the trends decreased by approximately
40 nT. After this, the trends were observed to increase by 20–50 nT over a 1–2-h interval. The amplitudes showed
fluctuations that did not exceed ±(3–4) nT.
*KAK Station.* On 13 January 2022, used as a quiet time reference, the trend $\overline{X}$ increased from –20 nT to 5 nT over
the 03:00–08:00 UTC period, and then gradually decreased from 5 nT to 0 nT (Figure 14). The trend $\overline{Y}$ decreased to –19 nT
by 05:00 UTC, after that it increased to 0 nT by 08:00 UTC and then remained essentially constant. The trend $\overline{Z}$ increased
from –15 nT to 8 nT over the 03:00–06:00 UTC period, next it decreased from 7 nT to 0 nT from 07:00 UTC to 09:00 UTC,
and then almost did not change. The magnitude of fluctuations in all components did not exceed ±1 nT. On 17 January 2022
used as a quiet time reference, the trend $\overline{X}$ increased from –10 nT to 5 nT, though a reduction in the dependence $\overline{X}(t)$ was
observed to occur from 08:00 UTC to 15:00 UTC, and $\overline{X} \approx 6$ nT from 15:00 UTC to 17:00 UTC. The trend $\overline{Y}$ first
decreased to –17 nT, and then increased to 5 nT. The trend $\overline{Z}$ increased to 2 nT by 06:40 UTC and dropped from 2 nT to –3
nT over the 06:40–12:00 UTC period. Then $\overline{Z} \approx 2$ nT. The amplitude showed fluctuations attaining 2–3 nT in every
component.
During the course of the day the volcanic explosion occurred, the magnitude of fluctuations in every component
increased noticeably. The trend $\overline{X}$, fluctuating, decreased from 6 nT to –7 nT from 06:00 UTC to 10:00 UTC. Next, it
increased from –7 nT to 10 nT over a 3-h interval, and finally the trend decreased to –13 nT. The trend $\overline{Y}$ also first
increased from –27 nT to –7 nT, then it decreased by less than 10 nT over a 1.5-h interval, after which the trend $\overline{Y}$ increased
to 17 nT at 13:30 UTC. A clear quasi-periodic perturbation with a period of $T \approx 55$–60 min and a strength of 4 nT was
recorded from approximately 11:00 UTC to 14:00 UTC. Other disturbances had amplitudes of 1–1.5 nT. The trend $\overline{Z}$
showed fluctuations within the ±3-nT limits throughout the 03:00–07:00 UTC period, when the trend remained almost the
same, next, from 07:00 UTC to 12:00 UTC, the trend decreased from 3 nT to –5 nT, and finally it increased. The fluctuations
occurred with an amplitude of ~1 nT.






Figure 13: Same as in Figure 2 but for the LRM Station.






**Figure 14: Same as in Figure 2 but for the KAK Station.**



*KNY Station.* During 13 January 2022, the trends $\overline{X}$ and $\overline{Z}$ increased from –21 nT to 5 nT from 03:00 UTC to
08:00 UTC (Figure 15), while the trend $\overline{Y}$ was observed to develop a deep drop from 4 nT to –25 nT. From 08:00 UTC to
17:00 UTC, the strength of fluctuations of the geomagnetic field was insignificant, ±1 nT. On 17 January 2022, a deep drop
occurred in the $\overline{Y}$ and $\overline{Z}$ trends. The magnitude of fluctuations in all components attained ±5 nT. On 15 January 2022, the
magnitudes of the trends in all components increased over the 03:00 UTC to 06:00 UTC period. The strength and time rate
of fluctuations also increased. Six groups of perturbation were observed to arrive with time delays of 39 min, 95 min, 140
min, 270 min, 435 min, and 685 min (see Figure 15); the amplitudes of the disturbances attained 4–5 nT. A pronounced *Y*-
component oscillation with $T \approx 70$-min period and an amplitude of 4-nT arrived with the time delay of 435 min.
*MMB Station.* On 13 January 2022, the trend of *X(t)* increased from –19 nT to 5 nT, and then fluctuated around 2–3
nT (Figure 16). The *Y* component showed a negative bay disturbance, with a strength reduction from –4 nT to –17 nT, which
persisted from 03:00 UTC to 08:00 UTC. Then, up to 17:00 UTC, an insignificant rise in this component strength was
observed to occur. The *Z*-component, instead, exhibited a positive bay disturbance over the 03:00 UTC to 09:00 UTC period.
On 17 January 2022, the strengths of the *X*- and *Y*-components showed variations attaining 10–15 nT, whereas the *Z*-
component variations did not exceed 5–6 nT.
On 15 January 2022, the bay reductions by 10 nT, 10 nT, and 3 nT in the strengths of the *X*, *Y*, and *Z* components,
respectively, were observed to occur with time delays of 200–225 min and to persist for 210 min to 290 min. In addition, the
amplitudes showed quasi-periodic disturbances with amplitudes of a few nanotesla and $T \approx 7$–20 min.
*SHU Station.* During 13 January 2022, the trend $\overline{X}$ decreased from 3 nT to –2 nT over ~03:30 UTC to 17:00 UTC
period (Figure 17), while the trend $\overline{Y}$, instead, increased from –4 nT to 3 nT. The trend $\overline{Z}$ decreased from 4 nT to –2 nT
over the 04:00 UTC to 10:00 UTC period and then gradually increased from –2 nT to 1 nT. All components showed
fluctuations with amplitudes not exceeding ~1 nT. On 17 January 2022, the trend $\overline{X}$ was first found to fluctuate within the
0–5-nT limits; after 10:00 UTC, some surges and drops attained 10–20 nT, and their durations did not exceed 1 h, whereas
the time variations in the *Y*- and *Z*-components showed significant, up to 10–20 nT, fluctuations.
On the day the volcanic explosion occurred, the trend $\overline{X}$ first decreased from 5 nT to –17 nT over the 06:00 UTC
to 09:00 UTC period, then it increased from –17 nT to 15 nT. A drop from 15 nT to –15 nT in the $\overline{X}(t)$ dependence was
observed to occur from 15:00 UTC to 17:00 UTC, while a clearly observed oscillation of the strength with 4-nT amplitude
and $T \approx 50$-min period persisted over the 11:00–13:30 UTC period. The trend $\overline{Y}$ increased from –10 nT to 18 nT over the
time interval from 04:00 UTC to 09:40 UTC, next its decrease to –35 nT continued to 15:00 UTC; finally, a ~50 nT surge in
the trend persisted for ~1.5 h. The trend $\overline{Z}$ decreased from 20 nT to 0 nT over the 04:00 UTC to 11:00 UTC period. From
11:00 UTC to 16:00 UTC, the $\overline{Z}(t)$ showed a drop from 0 nT to –20 nT, while a clearly observed oscillation with a 6–7-nT
amplitude and an ~80-min period lasted from 12:00 UTC to 17:00 UTC.




**Figure 15: Same as in Figure 2 but for the KNY Station.**







Figure 16: Same as in Figure 2 but for the MMB Station.







**Figure 17: Same as in Figure 2 but for the SHU Station.**



*DLT Station.* On 13 January 2022, all components showed insignificant (a few nanotesla) fluctuations in their
strengths (Figure 18) over the time period beyond the data gap. 17 January 2022, the trend $\overline{X}$ decreased from 36 nT to –17
nT from 04:00 UTC to 12:00 UTC, then it exhibited fluctuations within the 4–6-nT limits. The trend $\overline{Y}$ decreased from 17
nT to –7 nT after 03:30 UTC. Next its strength showed fluctuations within the ±(2–3)-nT limits. The trend $\overline{Z}$ remained
almost constant from 03:00 UTC to 06:00 UTC, then it increased from –25 nT to 10 nT, and after 09:00 UTC it again
remained almost constant.
Throughout the day the volcanic explosion occurred, all components showed considerable variations. The trend $\overline{X}$
experienced a bay reduction from ~22 nT to –25 nT over the 06:00–11:00 UTC period. Another drop in the $\overline{X}(t)$
dependence was observed to occur from 13:00 UTC to 17:00 UTC. The $\overline{Y}$ showed significant and long-lasting disturbances
from 04:00 UTC to 17:00 UTC. The trend $\overline{Z}$ increased from –9 nT to 4 nT over the 04:20–05:30 UTC period, next a drop
in $\overline{Z}$ from 4 nT to 2 nT occurred, which was followed by an increase from 2 nT to 17 nT. After that a steep fall in the trend
to 0 nT was first observed, and then a slow decrease from 0 nT to –5 nT. After around 15:30 UTC, the trend once again
showed noticeable variations (~5 nT).
*CKI Station.* On 13 January 2022 used as a quiet time reference, the trend $\overline{X}$ showed an insignificant rise to ~26
nT before approximately 05:00 UTC, after which the trend experienced a sharp fall from 26 nT to –10 nT and later was
followed by insignificant fluctuations in its magnitude (Figure 19). The trend $\overline{Y}$ first decreased from –3 nT to –13 nT over
the 04:00–06:00 UTC period, next, from 06:00 UTC to 10:00 UTC, it experienced a steep rise from –13 nT to 13 nT that was
changed by a gradual reduction in the trend to 0 nT at 17:00 UTC. The trend $\overline{Z}$ fell from 5 nT to –3.5 nT over the 03:30
UTC to 06:30 UTC period, whereas it showed two considerable surges, from –2 nT to 2 nT and from –2 nT to 1 nT, over the
09:00–12:00 UTC and 12:00–17:00 UTC periods, respectively. On 17 January 2022, used as a quiet time reference, the trend
$\overline{X} \approx 20$ nT from 03:00 UTC to 06:00 UTC. From 06:00 UTC to 10:00 UTC, it was observed to steeply fall from 20 nT to –
15 nT. Noticeable surges (by 10 nT to 15 nT) were observed over the 12:00 UTC to 16:00 UTC and 16:00 UTC to 17:00
UTC periods. The trend $\overline{Y}$ sharply increased from –40 nT to 10 nT from 03:00 UTC to 06:00 UTC, and then, fluctuating,
gradually decreased from 10 nT to –10 nT at 17:00 UTC. The trend $\overline{Z}$ increased from 0 nT to 15 nT over the 03:00–05:50
UTC period, after this it sharply decreased to –5 nT over a 3-h interval. After 09:00 UTC, the trend $\overline{Z}$ showed fluctuations
within the ±(2–3)-nT limits.






**Figure 18: Same as in Figure 2 but for the DLT station.**





**Figure 19: Same as in Figure 2 but for the CKI station.**





On the day the volcanic explosion occurred, all components showed a significant enhancement in their variations.
The trend $\overline{X}$ first increased to 05:30 UTC and then sharply decreased from 20 nT to –10 nT after 06:00 UTC. Noticeable
increases in $\overline{X}$ from –10 nT to 0 nT occurred during the 10:00–12:00 UTC period. From 13:00 UTC to 17:00 UTC, the
trend exhibited a drop from 0 nT to –27 nT. The trend $\overline{Y}$ increased from –5 nT to 10 nT from 04:00 UTC to 07:40 UTC.
Next, from 07:40 UTC to 12:30 UTC, $\overline{Y}$ experienced a bay reduction from ~8–10 nT to –8 nT. After 12:30 UTC, the trend
$\overline{Y}$ was observed to decrease to –14 nT at 17:00 UTC. The trend $\overline{Z}$ sharply decreased from 15 nT to –13 nT during the
03:30–09:30 UTC period. From 09:30 UTC to 15:00 UTC, the dependence $\overline{Z}(t)$ exhibited a surge from –13 nT to 5 nT. Yet
another surge in $\overline{Z}$ to 7 nT was observed to occur from 15:00 UTC to 17:00 UTC.
*GAN Station.* On 13 January 2022, the trend $\overline{X}$ first increased from –12 nT to 21 nT over the 03:00 UTC to 08:00
UTC period, and then decreased from 21 nT to –14 nT during the 08:00–17:00 UTC period (Figure 20). The trend $\overline{Y}$ first
increased from –1 nT to 5 nT from 03:00 UTC to 04:00 UTC, then sharply decreased from 5 nT to –25 nT from 04:00 UTC
to 08:45 UTC, next increased from –25 nT to 13 nT over the 09:00–12:00 UTC period, and finally gradually decreased from
13 nT to 8 nT at 17:00 UTC. The trend $\overline{Z}$ exhibited two considerable surges, from –4 nT to 5 nT over the 03:00 UTC to
06:00 UTC period, and from –4 nT to 11 nT from 08:30 UTC to 14:00 UTC. On 17 January 2022, the trend $\overline{X}$ first
increased from –10 nT to 40 nT from 03:00 UTC to 07:00 UTC, next decreased to –20 nT at 12:00 UTC, and then exhibited
fluctuations within the ±5-nT limits. The trend $\overline{Y}$ showed short-term (~1–2 h) increases by up to 4–5 nT, in addition to a
powerful surge (from –20 nT to 26 nT) during the 03:00–11:30 UTC period. The trend $\overline{Z}$ exhibited a powerful surge from –
3 nT to 27 nT from 03:00 UTC to 10:00 UTC; relatively small undulations of up to 5–6 nT were observed to occur after
10:00 UTC.
On the day the volcanic explosion occurred, all components exhibited noticeably enhanced variability. The trend
$\overline{X}$ first increased from –7 nT to 28 nT over the 03:00 UTC to 06:00 UTC period and then reduced from 28 nT to –3 nT
from 06:00 UTC to 10:30 UTC; in addition to variations within the ±5-nT limits, a drop from 5 nT to –28 nT was observed
to occur over the 13:00–16:30 UTC period. The trend $\overline{Y}$ also first increased from –20 nT to 7 nT over the 03:00–06:30 UTC
period, and then a deep drop occurred from 7–18 nT to –8 nT over the 06:30 UTC to 14:25 UTC period. Over the 14:25–
17:00 UTC period, the trend $\overline{Y}$ decreased from 18 nT to 2 nT. The trend $\overline{Z}$ first increased to 22 nT before 06:00 UTC.
Next, a deep drop (from 22 nT to –20 nT) followed over approximately 7 h. And finally, the moderate (up to 10–15 nT)
variations in $\overline{Z}$ were observed to occur. It should be noted that synchronous geomagnetic bay disturbances were observed to
occur uncertainly at this most distant recording station.






**Figure 20: Same as in Figure 2 but for the GAN station.**





**7 Statistical data analysis of the bay excursions in geomagnetic field strengths**
Table 4 shows the basic parameters of the bay disturbances of the geomagnetic field, viz., the magnitudes ΔX, ΔY, ΔZ, the
time delays τ, and the durations ΔT of the northward component, X, of the eastward component, Y, and of the vertical
component, Z, at nineteen stations. As can be seen from Table 4, the values of ΔX, ΔY, and ΔZ were most often negative
except for the data from the API and PPT stations, which were located at a distance, r, of 840 km and 2,730 km,
respectively, away from the volcano. The time delay showed a tendency to increase with increasing distance from the
volcano, and the duration of disturbances exhibited the same tendency as the time delay. Table 4 shows that the strength of
disturbances exhibits a tendency to decrease with increasing r.
Table 4. Basic parameters of bay disturbances in the geomagnetic field.

| Station | $\Delta X$ (nT) | $\tau_X$ (min) | $\Delta T_X$ (min) | $\Delta Y$ (nT) | $\tau_Y$ (min) | $\Delta T_Y$ (min) | $\Delta Z$ (nT) | $\tau_Z$ (min) | $\Delta T_Z$ (min) |
|---|---|---|---|---|---|---|---|---|---|
| API | 15 | 16 | 90 | 28 | 16 | 146 | −13 | 16 | 130 |
| PPT | 8 | 45 | 90 | 8 | 45 | 100 | | | |
| EYR | −40 | 50 | 120 | −25 | 50 | 120 | −15 | 50 | 120 |
| CNB | −20 | 100 | 120 | −50 | 60 | 178 | −15 | 60 | 120 |
| CTA | −18 | 105 | 130 | −63 | 60 | 150 | −30 | 60 | 150 |
| MCQ | −80 | 100 | 180 | −50 | 100 | 150 | −30 | 80 | 150 |
| HON | −10 | 75 | 180 | −5 | 75 | 180 | −2 | 75 | 180 |
| ASP | −15 | 100 | 230 | −50 | 75 | 240 | −15 | 75 | 240 |
| KDU | −10 | 110 | 210 | −30 | 110 | 210 | −15 | 110 | 200 |
| IPM | −10 | 110 | 220 | −8 | 110 | 220 | −2 | 100 | 220 |
| GNG | −15 | 120 | 270 | −50 | 120 | 270 | −30 | 120 | 270 |
| LRM | −10 | 140 | 270 | −10 | 140 | 265 | −5 | 140 | 265 |
| KAK | −10 | 180 | 270 | −8 | 165 | 260 | −8 | 165 | 300 |
| KNY | −10 | 115 | 270 | | | | −8 | 120 | 270 |
| MMB | −10 | 165 | 240 | −8 | 165 | 240 | −8 | 165 | 240 |
| SHU | −10 | 105 | 220 | −10 | 150 | 200 | −10 | 150 | 200 |
| DLT | −20 | 165 | 240 | −8 | 165 | 240 | | | |
| CKI | −12 | 170 | 240 | −15 | 175 | 250 | −10 | 165 | 240 |
| GAN | −10 | 240 | 240 | −8 | 240 | 240 | −15 | 240 | 240 |




438    Figure 21 presents scatter plots of time delay versus distance from the volcano, which reveal the following linear

439 dependences:

440 $\tau_X = 17.17r + 9.3$, $\sigma \approx 22.1$ min, $R^2 \approx 0.83$,  (1)

441 $\tau_Y = 19.93r - 10$, $\sigma \approx 12$ min, $R^2 \approx 0.96$,  (2)

442 $\tau_Z = 19.63r - 12$, $\sigma \approx 14.1$ min, $R^2 \approx 0.94$,  (3)

443 where distance is in Mm, time delay is in min, $\sigma$ is a root mean square error, $R^2$ is an adjusted coefficient of determination.

444    The individual points are fit with the following straight lines (Figure 22):

445 $\Delta T_X = 18.44r + 86.5$, $\sigma \approx 36.3$ min, $R^2 \approx 0.68$,  (4)

446 $\Delta T_Y = 14.29r + 115.6$, $\sigma \approx 34.5$ min, $R^2 \approx 0.60$,  (5)

447 $\Delta T_Z = 15.63r + 109.8$, $\sigma \approx 39.5$ min, $R^2 \approx 0.56$.  (6)

448 The relations (1) – (3) and (4) – (6) indicate that the time delay and the duration of disturbance indeed increase with distance

449 from the volcano. The formation of disturbance is close to root mean square deviations in time delays, i.e., to 12–22 min.





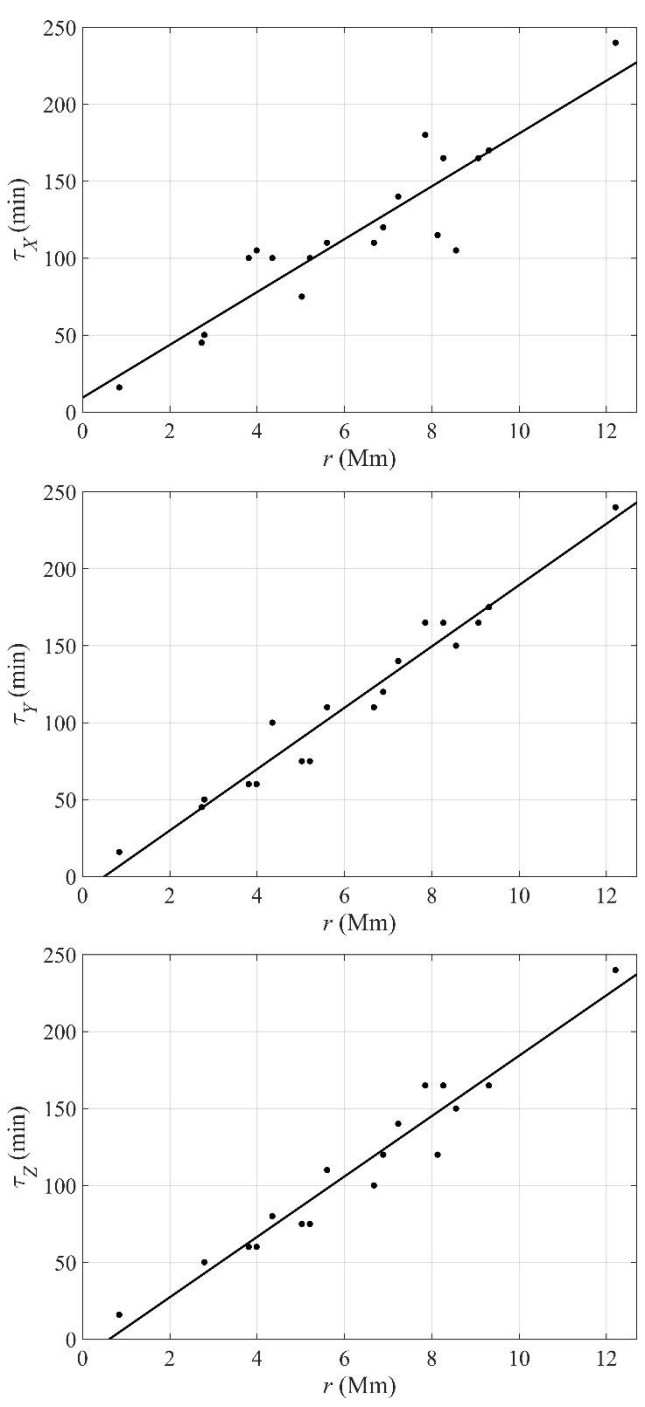

**Figure 21: Time delay of bay disturbance in the geomagnetic field vs distance, *r*, from the volcano and the estimated regression line superimposed on the scatter plot.**





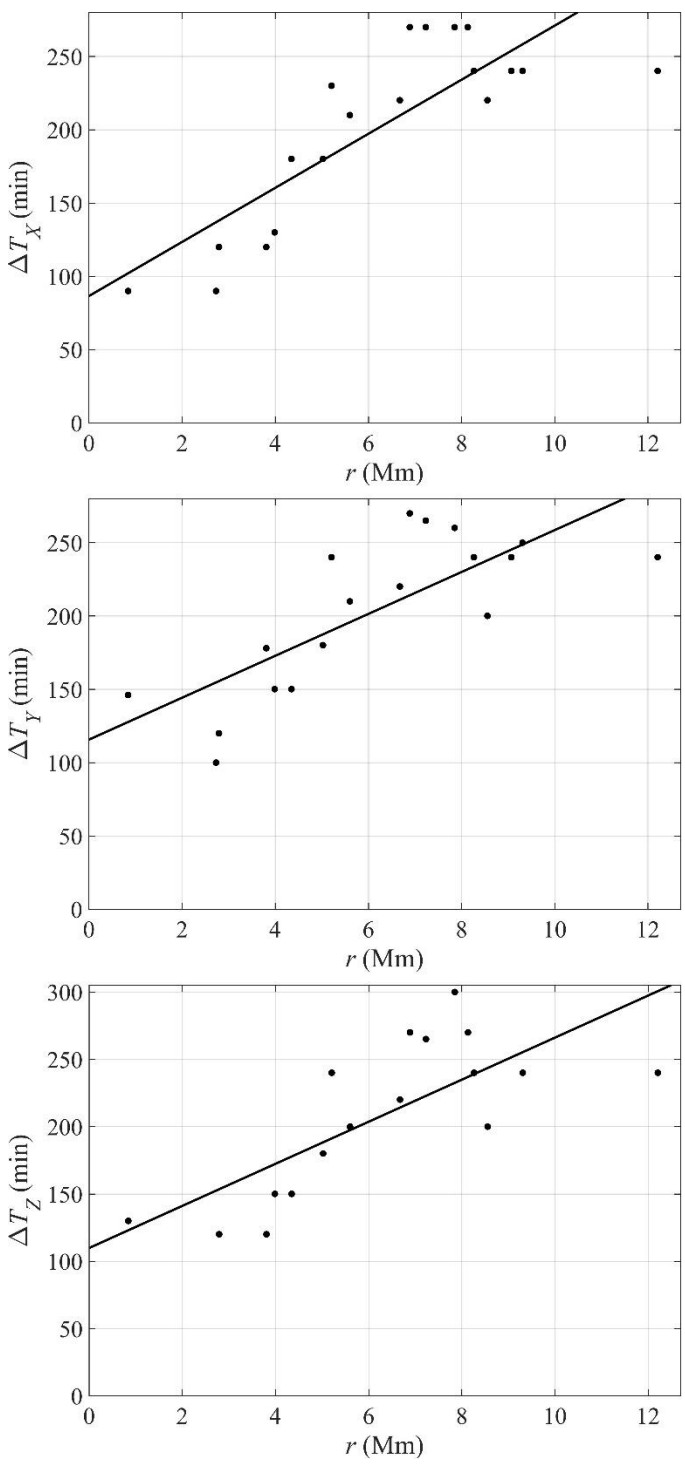

**Figure 22: Duration of bay disturbance in the geomagnetic field vs distance from the volcano and the estimated regression line superimposed on the scatter plot.**





Histograms showing the distributions of all $\Delta X$, $\Delta Y$, and $\Delta Z$ are presented in Figure 23. The most probable values of
these disturbances are seen to be as follows: for the northward component $\Delta X = -(9.0 \pm 5.1)$ nT, for the eastward component
$\Delta Y = -(10.5 \pm 5.6)$, and for the vertical component $\Delta Z = -(6.3 \pm 3.1)$ nT, and $-(25.0 \pm 5.0)$ nT.

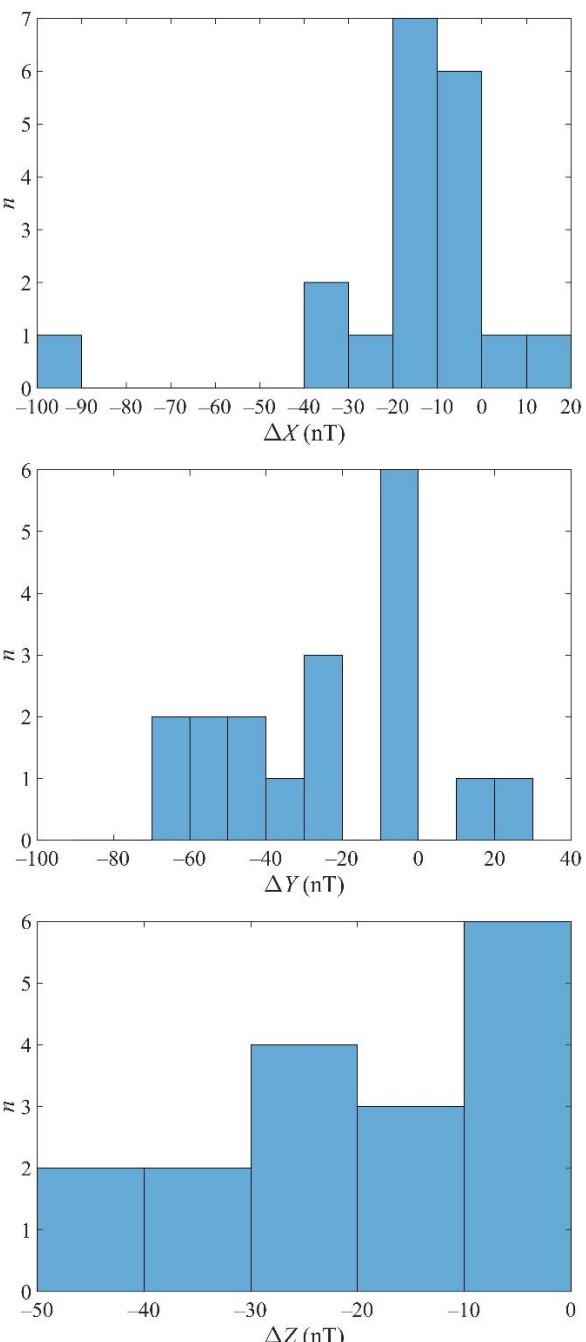


**Figure 23: Histogram showing the distribution of excursions in bay disturbances in the geomagnetic field.**





**8 Statistical data analysis of the quasi-periodic variations in geomagnetic field magnitudes**
The time delays of a possible response of the magnetic field to the volcanic explosion and the apparent speeds for six groups
of characteristic variations in the components of the geomagnetic field are presented in Table 3, which show that the
variations in the eastward component $Y$ are seen most clearly. Figure 24 presents a scatter plot of time delay versus distance
from the volcano for all the data presented in Table 3, which reveal the following linear dependences:
$\Delta t_1 = 4.157r + 5.1,$      $\sigma = 0.32$ min,     $R^2 = 0.9995,$                                           (7)
$\Delta t_2 = 11.14r + 4.6,$      $\sigma = 0.55$ min,     $R^2 = 0.9998,$                                           (8)
$\Delta t_3 = 16.66r + 4.6,$      $\sigma = 0.47$ min,     $R^2 = 0.9999,$                                           (9)
$\Delta t_4 = 33.13r + 4.6,$      $\sigma = 1.60$ min,     $R^2 = 0.9998,$                                          (10)
$\Delta t_5 = 53.11r + 6.1,$      $\sigma = 9.98$ min,     $R^2 = 0.9969,$                                          (11)
$\Delta t_6 = 82.97r + 7.7,$      $\sigma = 2.61$ min,     $R^2 = 0.9999.$                                          (12)
If $r \rightarrow 0$, then $\Delta t_0 \approx 4.6$–$7.7$ min. Such a time interval is needed for the wave to reach ionospheric heights, or more precisely,
$E$ region dynamo heights.

474          Use of relations in Eqs. (7) – (12) and the formula given by

$$v = \left( \frac{d\Delta t}{dr} \right)^{-1}$$
yields the following average speeds: $v_1 \approx 4$ km/s, $v_2 \approx 1.5$ km/s, $v_3 \approx 1$ km/s, $v_4 \approx 503$ m/s, $v_5 \approx 314$ m/s, and $v_6 \approx 209$ m/s.
These values are close to the values inferred from the histograms in Figure 25.

478          The horizontal apparent speed of propagation of disturbances can be estimated from the following relation:

$$v = \frac{r}{\Delta t - \Delta t_0}$$
where $\Delta t_0$ is the time taken for the blast wave to travel from the volcano to the $E$ region dynamo.





**Figure 24: Time delay of the onset of quasi-periodic disturbances in the geomagnetic field vs distance from the volcano and the estimated regression line superimposed on the scatter plot.**






**Figure 25: Histogram showing the distribution of the apparent speeds of propagation of quasi-periodic disturbances**

**in the geomagnetic field.**



### 9. Discussion

*Bay disturbances of the geomagnetic field.* During the day the Tonga volcanic explosion occurred, all three components usually exhibited geomagnetic bay disturbances whose absolute values were observed to be 10–60 nT, and the disturbances themselves were more often seen to be negative. The eastward component $Y$ experienced the largest disturbances, with the average value of –53 nT, whereas disturbances in the $X$ and $Z$ components were observed to be, on average, –15 nT. The smallest disturbance took place at the PPT station. The geomagnetic bay disturbances were virtually absent, or, more precisely, did not exceed –(2–8) nT also at the IPM station, which was located east of the volcano as well. This can be explained by the location of these stations on the night side of the Earth where the electron and electric current densities were approximately an order of magnitude smaller than in the sunlit ionosphere.

Disturbances were insignificant and unclear at the GAN station, the most distant station included in this study.

It should be stressed that the bay variations in the magnitudes of all geomagnetic field components did not exceed 5–10 nT during the days used as a quiet time reference. This observation supports the idea that the geomagnetic bay disturbances observed on 15 January 2022 were due to the volcanic explosion. However, this is a necessary but not sufficient condition for the volcanic explosion to be the cause of the effect.

A sufficient condition is a tendency for the time delay and duration of bay disturbance to grow with distance from the volcano, while a tendency for the disturbance strength was to decrease with distance from the volcano (see Figures 21 and 22).

The relations in Eqs. (1) – (3) suggest that, in the limit $r \rightarrow 0$, the minimum in the time delay, $\tau_{\min}$, is determined by root mean square error in the approximation, which is close to 14–22 min for the $X$, $Y$, and $Z$ components. The disturbance from the volcano takes such a time interval to travel from the volcano to the $E$ region dynamo, $z \approx 90–150$ km altitude, and to generated magnetic disturbance. The relations in Eqs. (1) – (3) permit estimates of the average speeds of the disturbances to be made using the relation given by

$$v = \left( \frac{d\tau}{dr} \right)^{-1}.$$

Then, $v_X \approx 970 \pm 235$ m/s, $v_Y \approx 836 \pm 103$ m/s and $v_Z \approx 849 \pm 121$ m/s. These magnitudes of the speeds are close to the blast wave speed [Chernogor, 2023b; Chernogor, 2023c].

It is important that the magnitudes of the speeds obtained are close to the speed of propagation of the disturbances in the electron density, $N$, and in the total electron content [Chernogor, 2023a]. This means that the formation of the ionospheric hole is the cause of the bay excursions in the geomagnetic field [Chernogor, 2023a].

Estimation of the magnitude of a bay disturbance in the geomagnetic field may be performed from the average daytime value of $N$ in the $E$ region dynamo of $(2–3) \times 10^{11}$ m$^{-3}$ and a neutral wind speed of $w \approx 100$ m/s. Then, the electric current density in the ionosphere is given by





$j_0 = eNw \approx (3.2{-}4.8) \times 10^{-6}$ A/m$^2$
where $e$ is the charge of an electron. The disturbance in $N$ within the ionospheric hole is estimated to be 5–20%, which yields
the perturbation in the ionospheric current of $\Delta j \approx (1.6{-}9.6) \times 10^{-7}$ A/m$^2$. The estimate of the disturbance in the magnetic
field follows from Maxwell's curl equation, given by
$\Delta B \approx \mu_0 \Delta j \Delta z$ (13)
where $\mu_0$ is magnetic permeability, $\Delta z \approx 50$ km is the thickness of the dynamo region. Substituting the numerical magnitudes
yields $\Delta B \approx 10{-}60$ nT, which is in excellent agreement with observations (~10–60 nT).
Thus, there is every reason to believe that the bay disturbances of the components of the geomagnetic field are
related to the generation of the ionospheric hole as a result of the explosion of Tonga volcano.
*The effect of atmospheric acoustic resonance*. The station nearest to Tonga volcano is the API Station. The *Y*
component exhibited the first perturbation over the 04:21–04:57 UTC period, i.e., the time delay was observed to be $\Delta t_0 \approx 6$
min. The acoustic wave take such a time interval to travel to the ionospheric *E* region where dynamo electric fields are
generated and where the generation of this magnetic effect occur. The sound wave is reflected at an altitude of $z_r \approx \overline{v}_x \Delta t_0 \approx$
110–120 km (where $\overline{v}_x \approx 300{-}330$ m/s is an average speed of sound), i.e., in the *E* region dynamo. It is important that the
period of the disturbance $T_0 \approx 4{-}4.5$ min and its duration $\Delta T_0 \approx 32{-}36$ min. These values indicate that the magnetic effect
have been generated by the atmospheric acoustic resonance in the Earth–*E* region dynamo cavity, where the volcanic
explosion excited the vibrations.
Since the API station is located at a range of ~840 km from the volcano, the radius, $r_L$, of the footprint of the
magnetic flux tube associated with the volcano is equal or greater than 1,000 km. This means that the magnetic effect from
the atmospheric acoustic resonance could be observed in the magnetically conjugate region. Indeed, oscillations with the
same period, $T_0$, duration $\Delta T_0$, and ~0.2-nT amplitude, were observed by [Iyemori et al., 2022; Yamazaki et al., 2022]. It is
important that the time delay was equal to $\Delta t_0 \approx 6$ min. This means that the disturbance from the API station was transferred
to the HON station along the magnetic flux tube ~10 Mm long at an Alfvén speed, $v_A$, of ~1 Mm/s for ~10 s, which much
shorter than $\Delta t_0$. It should be noted that the HON station is located about 900 km from the center of the magnetic flux tube,
and $r_L > 900$ km.
*Quasi-periodic disturbances*. Other disturbances with other time delays were superimposed on the disturbance due
to acoustic resonance (see Table 3). In total, the number of such disturbances could be six. Table 3 shows that six groups of
disturbances in the geomagnetic field also took place at other stations. It is important that the time delay increases with
distance from the volcano. The premise of requiring the time delays of the magnetic disturbances due to the volcanic
explosions to explain our observations is clearly supported by the INTERMAGNET Magnetometer Observatory data.
The values of the speeds were close to 4 km/s, 1.5 km/s, 1 km/s and 500 m/s, 313 m/s, and 200 m/s. All these
speeds have physical significance. The first and second group of speeds correspond to the speeds of the fast and slow MHD



waves [Sorokin and Fedorovich, 1982]. Approximately the same speeds were observed during powerful rocket launches
[Chernogor, 2009; Chernogor and Blaunstein, 2013]. The speed of $v_3 \approx 1$ km/s is characteristic of blast waves. This speed
was revealed after the Tonga volcanic explosion by [Matoza et al., 2022a; Matoza et al., 2022b]. The speed $v_4$ pertains to
atmospheric gravity waves at ionospheric heights [Chen et al., 2022; Themens et al., 2022]. Lamb waves that are generated
by massive releases of energy (exceeding 10 Mt of TNT) propagate at a speed of $v_5 \approx 313$ m/s over the Earth's surface
virtually without damping and partially penetrate to ionospheric heights along their propagation paths [Chernogor, 2022a;
Chernogor, 2022e; Chernogor, 2023a; Kubota et al., 2022; Lin et al., 2022; Zhang et al., 2022a]. The smallest speed of $v_6 \approx$
200 m/s probably pertains to an average speed of tsunami, which was observed after the volcanic eruption and generated
ionospheric disturbances [Carvajal et al., 2022; Ramírez-Herrera et al., 2022; Tanioka et al., 2022; Terry et al., 2022].

559       *Estimation of the quasi-periodic effects.*

The amplitude of quasi-periodic disturbances usually showed variations not exceeding 1–3 nT. Such disturbances were
generated by quasi-periodic disturbances arising in the electric current density at $E$ region dynamo heights from the action of
waves launched by the volcanic explosion.

563       The difference, $w_m$, in the drift velocities of ions and electrons, which are driven by the drag force of the neutral

atmosphere, causes the dynamo current density given by the relation
$j = eNw_m$.
The integrated in altitude current density is given by
$$J = \int_{\Delta z} j(z)dz\,.$$
Then, the amplitude of the quasi-periodic disturbance in the geomagnetic field is given by the following expression:
$\Delta B_a \approx \mu_0 J$.
If $N \approx (2\text{–}3) \times 10^{11}$ m$^{-3}$ on the sunlit side of the Earth, and $w_m \approx 0.3\text{–}1.5$ m/s, then $j \approx (1\text{–}7.2) \times 10^{-8}$ A/m$^2$, $J \approx (4.8\text{–}36) \times 10^{-}$
$^4$ A/m, and $\Delta B_a \approx 0.6\text{–}4.5$ nT. These estimates are seen to be close to magnitudes observed (~1–3 nT).

572       Thus, the disturbances in the geomagnetic field described above were observed on 15 January 2022 and were absent

during the days used as a quiet time reference. Consequently, they were most probably due to the volcanic eruption. These
disturbances were transported by the waves of various physical nature, viz., the fast and slow MHD waves, blast waves,
atmospheric gravity waves, Lamb waves, and ionospheric waves that arises from the tsunami.
**8 Conclusions**
Analysis of the data acquired at nineteen INTERMAGNET magnetic observatories revealed the following.





(1) During the day of the Tonga volcanic explosion, the variations in the magnitude of all components of the
geomagnetic field varied less monotonically than during the days used as a quiet time reference. The strength of fluctuations
also enhanced. All these factors indicated that the volcanic explosion led to the registered magnetic effect.
(2) The geomagnetic bay disturbances in all components of the geomagnetic field were observed to occur with a
time delay increasing with distance from the volcano from a few tens of minutes to 100–200 min. The magnitude of the
effect changed from ~10 nT to ~60 nT. The eastward component ($Y$) exhibited the greatest variations. The time delay and
duration of the disturbances increased with distance from the volcano, but amplitudes of the disturbances, instead, decreased.
The speed of propagation of the bay disturbances was close to the speed of the blast waves, approximately 700–1,000 m/s.
Geomagnetic bay disturbances were weakly expressed or were virtually absent on the night side of the planet. The premise
that the geomagnetic bay disturbances are closely related to the volcanic blast wave-induced formation of the ionospheric
hole has been validated.
(3) The quasi-periodic disturbances in the geomagnetic field arrived at the magnetic observatories with different
time delays. Six main groups of disturbances were identified. It is important that the time delay increases with distance from
the volcano in each group. The apparent speeds of propagation of the disturbances in each group have been estimated, and
the values of these speeds are as follows: 4 km/s, 1.5 km/s, 1 km/s and 500 m/s, 313 m/s, and 200 m/s. The first two speeds
pertain to the fast and slow MHD waves, the third to the blast wave, the fourth to the atmospheric gravity wave, the fifth to
the Lamb wave, and the six speed pertain to the tsunami.
(4) The magnetic effect due to the atmospheric acoustic resonance in the Earth – $E$ region dynamo cavity where
vibrations were excited by the volcanic explosion was observed at API Station, the nearest to Tonga volcano. The period of
the disturbance was estimated to be $T_0 \approx$ 4–4.5 min, the amplitude to be 2 nT, and its duration to be $\Delta T_0 \approx$ 32–36 min.
Similar effect was observed in the magnetically conjugate region at the HON station; however, its amplitude was an order of
magnitude smaller.
(5) Estimates of the bay and quasi-periodic disturbances are in good agreement with the parameters of disturbances
inferred from INTERMAGNET data.
**Competing interests**
The contact author has declared that none of the authors has any competing interests.
**Acknowledgments**
This publication makes use of data collected during the Tonga volcanic explosion by INTERMAGFNET and published at
https://www.intermagnet.org/. The solar wind parameters have been retrieved from the Goddard Space Flight Center Space
Physics Data Facility https://omniweb.gsfc.nasa.gov/form/dx1.html. This research also draws upon data provided by the





World Data Center for Geomagnetism, Kyoto (data are retrieved from http://wdc.kugi.kyoto-u.ac.jp). The author was
financially supported by the National Research Foundation of Ukraine (project 2020.02/0015, "Theoretical and experimental
studies of global disturbances from natural and technogenic sources in the Earth-atmosphere-ionosphere system") and by
Ukraine state-funded research projects #0121U109881 and #0121U109882. The author thanks Victor T. Rozumenko at V.
N. Karazin Kharkiv National University for helpful comments and his help in finalizing the manuscript. Special
acknowledgment is also due to Eugene G. Zhdanko for his assistance in graphic design.
**Data Availability Statement.**
The data sets discussed in this paper are freely accessible on the Internet at https://www.intermagnet.org.

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
