# Peer review of "A statistical study of the magnetic signatures of the unique Tonga volcanic explosion of 15 January 2022"

_Annales Geophysicae, 2023_

## Referee Comment (RC1)

**A statistical study of the magnetic signatures of the unique Tonga volcanic explosion of 15 January 2022**

Leonid F. Chernogor

Summary

This manuscript analyses geomagnetic effects of the 15 January 2022 eruption of the Hunga Tonga–Hunga Haʻapai volcano in the Tonga archipelago using data recorded at 19 nearby geomagnetic observatories.

The analysis identifies six small disturbances that are recorded in each observatories' X (North), Y (East), and Z (Vertical) time series and are assumed to be caused by the eruption, measuring:

1.  the time from the volcano eruption taken for each of the six "bay" disturbances to reach the observatory ($\Delta t_n$, n=1 to 6)
2.  the time from the volcano eruption taken for the most pronounced disturbance to reach the observatory ($\tau_X$, $\tau_Y$, $\tau_Z$) (Table 4)
3.  the peak deviation of the disturbances ($\Delta X$, $\Delta Y$, $\Delta Z$) (Table 4)
4.  the total length of time the six disturbances lasted in each component ($\Delta T_X$, $\Delta T_Y$, $\Delta T_Z$) (Table 4)

The paper then uses these measurements to calculate:

5.  the apparent speeds of these disturbances ($v'_n$, n=1 to 6) when travelling to each observatory (Table 3)
6.  the linear relationship between the arrival times of the most pronounced disturbances ($\tau_X$, $\tau_Y$, $\tau_Z$) and the observatories' distance from the volcano (Figure 21)
7.  the linear relationship between the duration of the disturbances ($\Delta T_X$, $\Delta T_Y$, $\Delta T_Z$) and the observatories' distance from the volcano (Figure 22)
8.  the most probable values of the peak disturbances ($\Delta X$, $\Delta Y$, $\Delta Z$) (Figure 23)
9.  the linear relationships between the arrival times of the six bay disturbances ($\Delta t_n$, n=1 to 6) and the observatories' distance from the volcano (Figure 24)
10. the time ($\Delta t_0$) taken for the disturbance generated by the volcano to reach ionospheric E-region heights
11. the distribution of apparent speeds ($v'_n$, n=1 to 6) of the bay disturbances (Figure 25)
12. the average speeds of the bay disturbances ($v_n$, n=1 to 6)

The calculations are interpreted to confirm that the disturbances observed in the observatory time series were caused by effects related to the volcanic eruption.

Comments

An assessment of the magnetic effects generated by the Hunga Tonga–Hunga Haʻapai volcano is a worthwhile endeavour. The author is to be congratulated on the careful and detailed description and analysis of a significant amount of information from 19 geomagnetic observatories.

Unfortunately, I could not understand what disturbances were being identified at the observatories on the day of the eruption. Figures 2 to 20 identify six disturbances in each in each of the magnetic-field components at each observatory, using the symbols $\Delta t_n$ (n=1 to 6) and arrows to point to the disturbance (see example figure below for PPT observatory). I could not see obvious disturbance features in the manuscript figures or in publicly available original data for the Australian observatories. The manuscript also refers to peak deviations and pronounced disturbances at the observatories, none of which were clearly identified. This lack of clarity about the basic data used in

later analysis is a significant concern and a serious flaw of the manuscript. It would be very helpful if the manuscript clearly showed and described each disturbance feature being identified and why that feature is considered likely to be a volcanic effect.

[Figure]

Figures 2 to 20 take up a lot of page space. It seems to me that the inclusion of plots for the quiet days 13 and 17 January, and their description in the text, does not add significant context to the analysis of the data of 15 January. I suggest consideration be given to omitting these parts of the figures and sections of the text. If higher-resolution plots of the 15 January observatory data assisted in identifying the disturbance features, perhaps this additional space would allow room for such plots.

The units Mm are not common. I suggest changing the units and related quantities to equivalent km.

Line 134 of the manuscript refers to the use of Fourier and wavelet transforms but there is no later evidence of the transformed data being used.

Consistently use either [] or () for parenthetical citations throughout the manuscript.

Corrections
Some details that should be addressed:

1. Line 42,43 (L 42,43): correct the citations format
2. L 53: change 14 to 19 stations
3. L 88: change volcanic to volcano

Conclusion
The investigation of magnetic-field effects related to the 2022 Tonga volcano eruption is worthwhile. However, the manuscript should clarify exactly what geomagnetic disturbance features are being identified in the observatory data. Measurements made about these features are the basis for all later analysis in the manuscript. This clarity will assist in understanding the veracity of the analysis. I suggest the manuscript be returned to the author requesting these clarifications.

---

## Referee Comment (RC2)

**A statistical study of the magnetic signatures of the unique Tonga volcanic explosion of 15 January 2022**

Leonid F. Chernogor

Dear Professor Chernogor

Thank you for your responses to the points raised in my earlier review.

Unfortunately, I am still unable to recognise the significance of the bay and higher-frequency disturbances in the magnetic-field record at each observatory that you identify as being caused by the volcanic eruption. They appear to be features common to an active geomagnetic field.

The manuscript attributes the disturbed magnetic field on 15 January 2022 to the volcanic eruption. (For example, lines 578 to 580:

*During the day of the Tonga volcanic explosion, the variations in the magnitude of all components of the geomagnetic field varied less monotonically than during the days used as a quiet time reference. The strength of fluctuations also enhanced. All these factors indicated that the volcanic explosion led to the registered magnetic effect.*)

However, data from Canberra geomagnetic observatory (CNB) show that active magnetic conditions began on 14 January, before the volcano erupted, and continued until around 17 January, well after the main eruption, suggesting a solar rather than volcanic origin of the activity evident during 15 January (see Figure 1 over page).

These disturbed activity levels are also reflected in the Kp index data (Table 1), which show that the higher activity began in the 18-21 3-hour period on 14 January and is correlated with Sunspot Numbers that increased to 98 on 14 January and remained elevated throughout the volcanic eruption period before returning to pre-eruption levels around 17 January.

Table 1. Kp index and Sunspot Number (SN) data from GFZ Potsdam for 13 to 20 January 2022.

| Date | 00-03 | 03-06 | 06-09 | 09-12 | 12-15 | 15-18 | 18-21 | 21-24 | SN |
|---|---|---|---|---|---|---|---|---|---|
| 13/01/2022 | 1 | 0 | 0 | 0.333 | 0.333 | 0.333 | 1 | 1 | 85 |
| 14/01/2022 | 0 | 0 | 0.333 | 1 | 2 | 2.667 | 4 | 5.667 | 98 |
| 15/01/2022 | 4.333 | 3.667 | 2.333 | 1.667 | 2.667 | 3 | 4 | 4.667 | 95 |
| 16/01/2022 | 3.667 | 3.333 | 2.333 | 2.667 | 2.333 | 2.667 | 4.333 | 3 | 102 |
| 17/01/2022 | 3 | 0.667 | 1 | 2 | 2.333 | 2.667 | 2.667 | 2.333 | 80 |
| 18/01/2022 | 5 | 3.333 | 3 | 2 | 2.333 | 3 | 2.667 | 3 | 66 |
| 19/01/2022 | 5.333 | 5 | 3.333 | 3 | 3 | 2 | 2 | 3 | 58 |
| 20/01/2022 | 3.333 | 0.333 | 0.667 | 0.333 | 0.333 | 0.667 | 0.333 | 0.333 | 57 |

This evidence suggests that solar activity was a key contributor to the geomagnetic-field disturbance levels observed during the period the manuscript analyses. This would make it more challenging, and all the more important, to clearly identify those features in the magnetic record that are caused by the eruption, distinguishing them from features of solar origin. I apologise if I am missing something obvious but I do not see that distinction in the manuscript.

Kind regards

Adrian Hitchman

[Figure]

Figure 1. Geomagnetic-field X, Y and Z components recorded at Canberra geomagnetic observatory (CNB) between 13 and 17 January 2022.

---

## Author Comment (AC4)

Dear Dr. Adrian Hitchman,

Thank you very much for your comment.

Regarding the disturbances in the magnetic-field record at each observatory that we identify as being caused by the volcanic eruption, they do appear to be features common to an active geomagnetic field. Moreover, these features do not have any specific appearance, and their appearances have nothing to do with the volcanic explosion.

The detection of the disturbances is based on revealing the disturbances, which have propagated with the same propagation speeds to all nineteen observatories. Altogether, six apparent speeds of 4 km/s, 1.5 km/s, 1 km/s, as well as 500 m/s, 313 m/s, and 200 m/s have been identified in a simultaneous analysis, for the first time.

The best evidence that the bay-shaped and quasi-periodic disturbances are caused by the action of the volcano is the dependence of the time delay on distance from the volcano. These dependences are already presented in Figures 21 and 24 in the manuscript.

Figure 21 shows the time delay of bay disturbance vs distance from the volcano and the estimated regression line superimposed on the scatter plot, while Figure 24 shows the time delay of the onset of quasi-periodic disturbances in the geomagnetic field vs distance from the volcano and the estimated regression line superimposed on the scatter plot.

The time delay vs distance from the volcano is also illustrated in the figure below, which we have constructed especially for you:

[Figure]

As an example, this figure shows UT variations in all 19 X-components of the geomagnetic field together, in the UT vs distance from the volcano plane. The vertical dashed line indicates the moment

of the volcanic explosion, while the six oblique straight regression lines virtually connect the possible moments of the onset of the magnetic field response indicated by the arrows in Figures 2–20. These variations have already been presented separately in Figures 2–20. Thus, these data clearly show that the disturbance time delay exhibits a tendency to increase with distance from the volcano, which testifies to the disturbance being propagated from the volcano. Moreover, we were able to establish that the bay-shaped disturbance of the geomagnetic field is associated with an ionospheric "hole" caused by a volcanic explosion and described, for example, in

Astafyeva, E., Maletckii, B., Mikesell, T. D., Munaibari, E., Ravanelli, M., Coisson, P., Manta, F., Rolland, L.: The 15 January 2022 Hunga Tonga eruption history as inferred from ionospheric observations, Geophysical Research Letters, 49 (10), e2022GL098827. https://doi.org/10.1029/2022GL098827, 2022.

Chernogor, L. F.: Ionospheric total electron content variations caused by the Tonga volcano explosion of January 15, 2022, Space Science and Technology, 29(3), 67-87, https://doi.org/10.15407/knit2023.03.067, 2023b.

The algorithm for finding the geomagnetic field response to the Tonga volcanic explosion is presented in the manuscript (Line 114–128), and the apparent speeds and the time delays found through applying the algorithm are collected in Table 3 (Line 158).

Other aspects of this study include the following.

1. Before searching for volcano effects, I carefully analyzed the state of space weather, for which I have developed a special format (see Fig. 1 at the end of this reply). Fig. 1 shows that a magnetic storm with Kp = 6– ≈ 5.667 occurred on January 14, 2022. From 00:00 UTC to 03:00 UTC on January 15, 2022, the Kp-index decreased to 4+ ≈ 4.333. Within the time interval of interest, approximately from 05:00 UTC to 18:00 UTC, the Kp values varied within the 1.667–3 range, i.e., there was no magnetic storm, the magnetic field was only slightly disturbed. On the reference days, Kp ≈ 0.333–1 (January 13, 2022), and Kp ≈ 0.667–2.333 (January 17, 2022). January 13, 2022, was ideal as a quiet time reference. Solar activity on January 15 was 10–15 units higher than on the reference days, however, this could only affect the trend level, but not the bay-shaped disturbance or quasiperiodic disturbances of the magnetic field.

2. A simple comparison of the temporal variations on January 13 and 15 shows that on January 13 the variations were smooth, and their amplitude did not exceed 1 nT (see, e.g., Figure 2 in the manuscript). On January 15, the magnitude and frequency of fluctuations increased significantly. Their amplitude was 1–3 nT.

3. The review of the literature on the geomagnetic field perturbations from the volcanic explosion is presented in the Introduction section (Line 44–73) of the manuscript, which I copy here for your convenience:

"Sun et al. (2022b) have estimated disturbances in the electric current in the ionospheric $E$ region caused by the Tonga volcanic explosion by making use of the data on geomagnetic field variations acquired by the global network of magnetometers. The $E$-region current density was estimated to be $J \approx 22–55$ mA/m$^2$ within a radius of 8,000 km away from the eruption, which changed the eastward components, $Y$, of the geomagnetic field by ~20–50 nT. The leading front of the disturbance traveled with a propagation speed of ~740 m/s. Le et al. (2022) investigated the effect that the volcano had on the equatorial electrojet and revealed the reversal of the electrojet direction due to a strong eastward zonal wind.

The explosion was also accompanied by variations in the geomagnetic field (Adushkin et al., 2022; Chernogor, 2023c; Chernogor and Holub, 2023a, 2023b; Iyemori et al., 2022; Le et al., 2022; Schnepf et al., 2022; Soares et al., 2022; Yamazaki et al., 2022). Adushkin et al. (2022) have described waves and disturbances in the atmospheric electric and magnetic fields. The data collected at 14 stations in the global network of observatories, INTERMAGNET, which are located in the 2.790–

6.225 Mm distance range from the volcano, have been used for investigating the magnetic effect. The disturbances in the geomagnetic field have been deduced to occur on a global scale, and two groups of disturbance have been revealed. In the first group, the disturbances were virtually synchronously observed immediately after the explosion, whereas in the second group, the magnetic disturbances appeared after the arrival of Lamb waves. Soares et al. (2022) described quasi-periodic disturbances in the magnitude of the eastward component, $Y$, with amplitude of ~3 nT and an ~4-min period observed with onset time delay of 10 min at 835-km distance from the volcano. The geomagnetic variations at 3.8-mHz (period of $T \approx 4.4$ min) have been analyzed by (Iyemori et al., 2022; Yamazaki et al., 2022), who relate these variations to the acoustic resonance. It is important to note that the oscillations at 3.8 mHz were observed simultaneously both in the vicinity of the volcano (API station) and in the magnetically conjugate region (HON station). The amplitudes of these virtually synchronous oscillations were observed to be 2 nT and 0.2 nT, respectively, while the time delay of the magnetic effect did not exceed 6 min. However, analogous oscillations were not observed at distances, $r$, greater than 2.7 Mm. The study by Schnepf et al. (2022) is concerned with the investigation of geomagnetic variations in the 3–8-min period range with amplitude of ~1 nT that were observed with a time delay of ~30 min (propagation speed of ~470 m/s). The authors relate these variations to the ionospheric wave, which was generated by the volcano, and explain the variations in the 13–93- and 5–100-min period ranges by the effects of tsunami and of atmospheric and ionospheric sources. Harding et al. (2022) describe the multi-instrument studies of the magnetic effect of Tonga volcano. They utilized the data collected by magnetometers at the ground and onboard the ICON and Swarm spacecraft to study the effect that the volcanic explosion had on neutral winds and the ionospheric dynamo current system on a global scale. Despite significant progress made in understanding the geomagnetic field disturbances related to the Tonga volcanic explosion, a further statistical and spectral analyses of these variations is to advance understanding of this scientific issue."

Further, I present excerpts from the papers, which had already been published before the study described in the manuscript. They illustrate individual elements of the geomagnetic effect of the Tonga volcanic explosion, as follows:

The conclusions arrived at the study by Schnepf, N. R., Minami, T., Toh, H., and Nair, M. C.: Magnetic Signatures of the 15 January 2022 Hunga Tonga–Hunga Ha'apai Volcanic Eruption, Geophysical Research Letters, 49 (10), e2022GL098454, https://doi.org/10.1029/2022GL098454 , 2022 are of interest to the current study with respect to characterizing disturbed geomagnetic conditions:

**4. Conclusions and Outlook**

15 January 2022 started and ended with disturbed geomagnetic conditions but conditions were relatively quiet around the time of the Hunga Tonga–Hunga Ha'apai eruption and stayed quiet through to when oceanic and atmospheric waves from the explosion reached the various Pacific geomagnetic observatories.

The local magnetic signature at API had periods of 3–8 min and strengths of ~1 nT arrived starting at 04:44 UTC and persisting until 05:38 UTC. The high frequency signature was visible in both API's vertical and horizontal components, suggesting an ionospheric origin. However, oceanic signals could be at play here and more work is needed to definitively separate the sources.

For Chichijima Island (CBI, Japan) and Easter Island (IPM, Chile), the local magnetic signals were concurrent with the eruption's water wave arrivals. At CBI, the magnetic signatures had period bands of 13–19 min (with corresponding amplitudes of 0.4–0.7 nT) and 49–93 min (with corresponding amplitudes of 1.8–2.4 nT). Meanwhile, at IPM, we identified magnetic signatures of 5–100+ min periodicity and 5–14 nT amplitude. It is unclear whether the signals at CBI and IPM are due to the eruption's tsunami water wave, deformation of the sea surface from atmospheric acoustic waves, ionospheric waves, or combinations of all these eruption-induced sources.

The Honolulu (HON) and Tahiti (PPT) observatories lacked clear magnetic signals concurrent with their island's water wave arrival time. Instead, similar to the other more inland observatories used in this study, recurrent magnetic signals were seen for the bulk of January 15th. These signals must be external in origin, however, it is ambiguous if they are related to the Hunga Tonga–Hunga Ha'apai eruption or to Earth's space weather conditions.

Future studies should pursue methods that separate internal and external magnetic field sources at each of the near-sea observatories. Additionally, incorporating atmospheric pressure data or ionospheric total electron content data could help distinguish the different sources creating the identified magnetic signatures. Numerical studies may also shed light in separating the magnetic signal from the tsunami water wave and the ionospheric disturbances. With such future work, we believe that the magnetic signatures from submarine volcanic eruptions can be rendered sensible.

The study by Adushkin, V. V., Rybnov, Y. S., and Spivak, A. A.: Wave-Related, Electrical, and Magnetic Effects Due to the January 15, 2022 Catastrophic Eruption of Hunga Tonga–Hunga Ha'apai Volcano, J. Volcanolog. Seismol., 16 (4), 251–263. https://doi.org/10.1134/S0742046322040029 , 2022. deals with the observations of perturbations in the atmosphere and in the geomagnetic field at global-scale distances from the volcanic explosion. The following excerpts from this paper are of interest (marked in yellow):
* * *
the period between ~04:10 and ~05:00 UTC in the shape of sign-varying variations with a period of ~60 s and peak amplitude ~20 V/m.

The results of instrumental observations show that, along with the explosion, the anomalous variations in the electrical field were also caused by wave disturbances of direct and antipodal origin. Figure 10 demonstrates variations in $E$ due to the arrival of the larger signals $P_1-P_4$ at the GMC. In particular, it follows from Fig. 10a that the arrival of the primary signal $P_1$ (arrival time ~18:25 UTC) gave rise to well-pronounced sign-varying variations in $E$ with a period of ~8 min and peak amplitude ~40 V/m. The variations in $E$ corresponding to the arrival of signals $P_2-P_4$ at the GMC are displayed in Figs. 10b–10d, respectively. The characteristics of electrical variations due to the arrival of signals $P_1-P_4$ are given in Table 3 as the amplitudes relative to the trend E* and period $T$. It should be noted that the sign-varying $E$ variations

~10 V/m.

**THE GEOMAGNETIC EFFECT OF THE VOLCANIC ERUPTION**

It is known that violent volcanic activity gives rise to increased variations in the Earth's magnetic field (Johnston, 1997; Spivak et al, 2020). The results of the present study also provide evidence that the explosion

**Table 3.** Characteristics of electrical variations during arrivals of atmospheric signals $P_1-P_4$ at the GMC

| Signal | Parameters | |
| --- | --- | --- |
| | $T$, min | $E^*$, V/m |
| $P_1$ | ~8 | ~40 |
| $P_2$ | ~4 | ~20 |
| $P_3$ | ~20 | ~20 |
| $P_4$ | ~6 | ~10 |

of Hunga Tonga–Hunga Ha'apai Volcano was accompanied by anomalous geomagnetic variations that occurred at great distances from the volcano. As an illustration, Figs. 11 and 12 show observations of the horizontal magnetic component (which is the most sensitive to external disturbances) $B_H = \sqrt{B_x^2 + B_y^2}$, made at the INTERMAGNET observatories at different distances from the volcano in the east–west and north–south directions, respectively (see Fig. 1). Inspection of Figs. 11 and 12 tells us that there is a well-pronounced change in the behavior of $B_H$ during the explosion in the shape of sign-varying variations whose duration reached ~60 min. We note that the anomalous variations were observed practically simultaneously at very different epicentral distances from the volcano, thus showing that the excited disturbance was global in character.

According to Spivak et al. (2020), geomagnetic variations were also observed when atmospheric signals arrived at recording sites. We will consider the geomagnetic effect that accompanied the signals $P_1$–$P_6$ using the MHV data. Figure 13 shows the geomagnetic variations at MHV that were recorded both during the volcanic explosion and when atmospheric signals arrived at the MHV. It should be noted that, overall, the variations in $B_H$ were recorded exactly during the explosion period and during the periods when atmospheric signals arrived.

The observed advance or delay in the geomagnetic variations relative to the times of arrival of the atmospheric signals can probably be explained by geophysical conditions, both along the propagation paths and at the recording sites.

[Figure]

**Fig. 11.** Variations in the horizontal component of the Earth's magnetic field during the January 15, 2022 explosion at Hunga Tonga–Hunga Ha'apai (the records were made at INTERMAGNET observatories situated east–west relative to the volcano); the epicentral distance is shown in the figures themselves (vertical arrows mark the explosion time).

[Figure]

**Fig. 12.** Variations in the horizontal component of the Earth's magnetic field during the January 15, 2022 explosion of Hunga Tonga–Hunga Ha'apai (the records were made at the INTERMAGNET observatories situated east and west of the volcano); the epicentral distances are shown in the figures themselves (vertical arrows mark the explosion time).

ones. That is to say, the atmospheric signal has traveled thrice around the globe, thus showing that the source energy was substantially above the value 50 Mt.

At the same time, we can also find the estimate from below for $W$ in application to the Hunga Tonga–Hunga Ha'apai eruption using data on the explosive eruption of Bezymianny Volcano (March 30, 1956) as reported in (Pasechnik, 1958; Pasechnik and Fedoseenko, 1958). The spectrum of the atmospheric signal due to the Bezymianny explosion is shown in Fig. 14. The value of $f_0$ is ~0.003 Hz for this case. The energy of the Bezymianny explosion as found in (Pasechnik and Fedoseenko, 1958) is $W \sim\approx 10^{16}$ J, or ~2.4 Mt of TNT. The estimates based on (1) gave $W \sim 3.8 \times 10^{16}$ J, or ~9 Mt, which is ~3.5 times the value reported in (Pasechnik and Fedoseenko, 1958). Bearing this in mind, we find that the estimate from below for $W$ in the explosion of Hunga Tonga–Hunga Ha'apai Volcano can amount to $W \sim 2.6 \times 10^{17}$ J, or ~60 Mt of TNT.

Further, it should be noted that it is not entirely clear at present what is the mechanism responsible for effects of volcanic eruptions on the Earth's magnetic and electrical fields. Local effects can apparently be attributed to intensive discharges of hot material into the atmosphere. However, the "long-range action" of volcanic explosions as found in the present study requires further more detailed research. It can be hypothesized that, as water–ash–gas mixture is violently emitted during the explosive phase of an eruption, a source of strong acoustic and electrical excitation acting on the ionosphere is being formed in the near-ground zone of the Earth. The result is to produce a magnetohydrodynamic disturbance at the epicenter of the source; the disturbance propagates at a great speed in the ionosphere (e.g., ~22 km/s (Sorokin and Fedorovich, 1982)).

The results of this study show that the air waves excited by the activity of a volcano, both direct and antipodal waves, also produce disturbances in the

It is also important to mention that this volcanic explosion produced significant variations in electrical and magnetic fields at considerable distances from the source of the disturbances. As well, the variations in the geophysical fields considered here were observed, not only during the explosion itself, but also when the atmospheric signals were arriving at recording sites.

It is difficult at present to offer a distinct physical interpretation for these effects. This problem requires further data acquisition and detailed analyses of the data. Also, it is necessary to develop analytical and calculable models of the process based on concrete mechanisms responsible for the action of volcanic explosion on the medium.

In our opinion, the above results provide a supplement to the relevant data base, and can be of interest for improving the existing models and developing new models to describe the action of volcanic activity on the geophysical medium and their verification.

**FUNDING**

This work was supported by the state assignment no. 1021052706233-4-1.5.4 "The Occurrence of Natural and Manmade Processes in Geophysical Fields (FMWN-2022-0012)".

**REFERENCES**

Adushkin, A.A. and Firstov, P.P., Explosive processes of volcanic eruptions and their manifestations in wave disturbances in the atmosphere, in *Ekstremalnye prirodnye yavleniya i katastrofy* (Extreme Natural Occurrences and Disasters), vol. 2, Moscow: IFZ RAN, 2010, pp. 264–278.

Adushkin, V.V. and Spivak, A.A., Impact of natural extreme events on geophysical fields in the environment, *Izvestiya, Physics of the Solid Earth*, 2021, vol. 57, no. 5, pp. 583–592.

Adushkin, V.V., Gostintsev, Yu.A., and Firstov, P.P., On the origin of air waves due to strong explosive eruptions, *Vulkanol. Seismol.*, 1984, no. 5, pp. 3–11.

The study presented in the paper by Soares, G., Yamazaki, Y., and Matzka, J.: Localized geomagnetic disturbance due to ionospheric response to the Hunga Tonga eruption on January 15, 2022, Geophysical Research Letters, https://doi.org/10.1002/essoar.10510482.1 , 2022 reaches the conclusion that ionospheric currents are the likely cause of the geomagnetic disturbance at Apia:

19 **Abstract**

20 The Hunga Tonga-Hunga Ha'apai volcano in the Pacific Ocean erupted on January 15, 2022. The
21 energy released by this submarine eruption caused waves propagating through the lithosphere,
22 ocean and atmosphere. Less than 10 minutes after the eruption, pulsation-like geomagnetic
23 disturbances started at the geomagnetic observatory Apia, approximately 835 km from Hunga
24 Tonga, and lasted for about 2 hours. These disturbances were most prominent in the Y (east)
25 component, with an oscillation amplitude of ~3 nT and dominant periods of 276, 254 and 219 s.
26 Comparable geomagnetic disturbances are absent at neighboring as well as high-latitude
27 geomagnetic observatories, indicating that the disturbances are localized and not related to solar
28 wind energy input. Tide gauge data show that tsunami waves arrived at Apia more than one hour
29 after the eruption. This leaves ionospheric currents as the likely cause of the geomagnetic
30 disturbances.

The study by Yamazaki, Y., Soares, G., and Matzka, J.: Geomagnetic Detection of the Atmospheric Acoustic Resonance at 3.8 mHz During the Hunga Tonga Eruption Event on 15 January 2022, Journal of Geophysical Research: Space Physics, 127 (7), e2022JA030540, https://doi.org/10.1029/2022JA030540 , 2022 arrives at the conclusion that the geomagnetic variation at Apia is most likely due to ionospheric dynamo currents driven by the acoustic resonance of the atmosphere:

**JGR Space Physics**

**RESEARCH ARTICLE**

10.1029/2022JA030540

**Key Points:**
- The effect of the January 2022 Hunga Tonga-Hunga Ha'apai volcano eruption on the geomagnetic field is examined
- Geomagnetic oscillation with a frequency of ~3.8 mHz is observed simultaneously near the volcano and its magnetic conjugate point
- The oscillation is attributed to the acoustic resonance of the atmosphere

**Correspondence to:**
Y. Yamazaki,
yamazaki@iap-kborn.de

**Citation:**
Yamazaki, Y., Soares, G., & Matzka, J. (2022). Geomagnetic detection of the atmospheric acoustic resonance at 3.8 mHz during the Hunga Tonga eruption event on 15 January 2022. *Journal of Geophysical Research: Space Physics*, 127, e2022JA030540. https://doi.org/10.1029/2022JA030540

Received 8 APR 2022
Accepted 22 JUN 2022

**Geomagnetic Detection of the Atmospheric Acoustic Resonance at 3.8 mHz During the Hunga Tonga Eruption Event on 15 January 2022**

Yosuke Yamazaki[1] , Gabriel Soares[2] , and Jürgen Matzka[3]

[1]Leibniz Institute of Atmospheric Physics at the University of Rostock, Kühlungsborn, Germany, [2]Observatório Nacional, Rio de Janeiro, Brazil, [3]GFZ German Research Centre for Geosciences, Potsdam, Germany

**Abstract** Modeling studies have predicted that the acoustic resonance of the atmosphere during geophysical events such as earthquakes and volcanos can lead to an oscillation of the geomagnetic field with a frequency of about 4 mHz. However, observational evidence is still limited due to scarcity of suitable events. On 15 January 2022, the submarine volcano Hunga Tonga-Hunga Ha'apai (20.5°S, 175.4°W, Tonga) erupted in the Pacific Ocean and caused severe atmospheric disturbance, providing an opportunity to investigate geomagnetic effects associated with acoustic resonance. Following the eruption, geomagnetic oscillation is observed at Apia, approximately 835 km from Hunga Tonga, mainly in the Pc 5 band (150–600 s, or 1.7–6.7 mHz) lasting for about 2 hr. The dominant frequency of the oscillation is 3.8 mHz, which is consistent with the frequency of the atmospheric oscillation due to acoustic resonance. The oscillation is most prominent in the eastward (Y) component, with an amplitude of ~3 nT, which is much larger than those previously reported for other events (<1 nT). Comparably large oscillation is not found at other stations located further away (>2700 km). However, geomagnetic oscillation with a much smaller amplitude (~0.3 nT) is observed at Honolulu, which is located near the magnetic conjugate point of Hunga Tonga, in a similar wave form as at Apia, indicating interhemispheric coupling. This is the first time that geomagnetic oscillations due to the atmospheric acoustic resonance are simultaneously detected at magnetic conjugate points.

**1. Introduction**

The paper by Iyemori, T., Nishioka, M., Otsuka, Y., et al.: A confirmation of vertical acoustic resonance and field-aligned current generation just after the 2022 Hunga Tonga Hunga Ha'apai volcanic eruption, Earth Planets Space, 74, 103, https://doi.org/10.1186/s40623-022-01653-y , 2022 examines the geomagnetic oscillations at Apia and Honolulu caused by the volcanic explosion in detail. We copied below only three excerpts from this paper:

**A confirmation of vertical acoustic resonance and field-aligned current generation just after the 2022 Hunga Tonga Hunga Ha'apai volcanic eruption**

Check for updates

Toshihiko Iyemori[1*] [iD], Michi Nishioka[2], Yuichi Otsuka[3] and Atsuki Shinbori[3]

**Abstract**

A strong volcanic eruption caused a clear vertical acoustic resonance between the sea surface and the thermosphere. Its effects are observed as geomagnetic and GPS-TEC oscillations near the volcano and its geomagnetic conjugate area. The geomagnetic oscillations are observed at Apia and Honolulu geomagnetic observatories with amplitude of about 2 nT and 0.2 nT, respectively. The volcanic eruption started around 04:14 UT on January 15, 2022. The oscillations appeared at 04:21UT at Apia, Samoa, only about 7 min after the start of eruption. Because the distance between the volcano and Apia is about 841 km, it takes about 40 min for a sound wave to propagate from the volcano to Apia. Therefore, it is more plausible to assume that the magnetic oscillation observed at Apia about 7 min after the eruption is caused by the sound waves propagated vertically upward to the ionosphere and generated an electric current. The coherent appearance of geomagnetic oscillation at Honolulu located near the geomagnetic conjugate point of the volcano strongly support the idea that the ionospheric current generated over the volcano diverted as a field-aligned current which flew to the opposite hemisphere and caused the geomagnetic oscillation at Honolulu. The earliest start of GPS-TEC oscillation was around 04:15UT near the volcanic eruption, and it was around 04:20 UT at KOKV station in Hawaii. The time-lag of the TEC variations between Samoa and Hawaii obtained by a cross-correlation analysis is 4.5 min or 8.5 min. These time differences are much smaller than the travel time of the seismic waves from the

The statement above (marked in blue) is confirmed by the entire paper, while Figures 2 and 10 below present the data:

[Figure]

**Fig. 2** Enlarged plots of high-pass filtered geomagnetic components. An oscillation at Apia start around 04:21 UT, about 7 min after the start of eruption. Although the amplitude is small, coherent oscillations with those in Apia encircled by orange and green dotted lines are observed at Honolulu

[Figure]

**Fig. 10** Phase relation of magnetic field oscillations between Apia and Honolulu

The study by Le, G., Liu, G., Yizengaw, E., and Englert, C. R.: Intense equatorial electrojet and counter electrojet caused by the 15 January 2022 Tonga volcanic eruption: Space- and ground-based observations, Geophysical Research Letters, 49 (11), e2022GL099002, https://doi.org/10.1029/2022GL099002 , 2022 **presents an analysis indicating that the geomagnetic storm had a minimal impact on dayside equatorial electrodynamics**:

[Figure]

[Figure]

**Geophysical Research Letters**

**RESEARCH LETTER**

10.1029/2022GL099002

**Key Points:**

- Space- and ground-based observations reveal dramatic equatorial electrojet variations caused by the Tonga volcanic eruption
- Strong eastward turning of atmospheric zonal winds in the E-region is responsible for the directional reversal of the equatorial electrojet
- The observed complex spatiotemporal variations can be explained by a large-scale disturbance propagating eastward from the eruption site

**Correspondence to:**

G. Le,
Guan.Le@nasa.gov

**Citation:**

Le, G., Liu, G., Yizengaw, E., & Englert, C. R. (2022). Intense equatorial electrojet and counter electrojet caused by the 15 January 2022 Tonga volcanic eruption: Space- and ground-based

**Intense Equatorial Electrojet and Counter Electrojet Caused by the 15 January 2022 Tonga Volcanic Eruption: Space- and Ground-Based Observations**

Guan Le[1] , Guiping Liu[1,2,3] , Endawoke Yizengaw[4] , and Christoph R. Englert[5]

[1]ITM Physics Laboratory, Heliophysics Division, NASA Goddard Space Flight Center, Greenbelt, MD, USA, [2]The Catholic University of America, Washington, DC, USA, [3]Space Sciences Laboratory, University of California, Berkeley, CA, USA, [4]Space Science Application Laboratory, The Aerospace Corporation, El Segundo, CA, USA, [5]Space Science Division, U.S. Naval Research Laboratory, Washington, DC, USA

**Abstract** We present space- and ground-based multi-instrument observations demonstrating the impact of the 2022 Tonga volcanic eruption on dayside equatorial electrodynamics. A strong counter electrojet (CEJ) was observed by Swarm and ground-based magnetometers on 15 January after the Tonga eruption and during the recovery phase of a moderate geomagnetic storm. Swarm also observed an enhanced equatorial electrojet (EEJ) preceding the CEJ in the previous orbit. The observed EEJ and CEJ exhibited complex spatiotemporal variations. We combine them with the Ionospheric Connection Explorer neutral wind measurements to disentangle the potential mechanisms. Our analysis indicates that the geomagnetic storm had minimal impact; instead, a large-scale atmospheric disturbance propagating eastward from the Tonga eruption site was the most likely driver for the observed intensification and directional reversal of the equatorial electrojet. The CEJ was associated with strong eastward zonal winds in the E-region ionosphere, as a direct response to the lower atmosphere forcing.

**Plain Language Summary**

Thus, our results have significantly complemented the results obtained by the authors of the papers listed above.

The author is grateful to Dr. Adrian Hitchman for the thorough and comprehensive review of the manuscript.

Sincerely,

Leonid Chernogor.

[Figure]

**Fig. 1.** UT variations in the solar wind parameters: measured concentration, $n_{sw}$, of particles, temperature $T_{sw}$, radial velocity $V_{sw}$, calculated dynamic pressure $p_{sw}$, measured $B_z$ and $B_y$ components of the interplanetary magnetic field; calculated values of the energy, $\varepsilon_A$, transferred from the solar wind into the Earth's magnetosphere per until time; $K_p$-index and $D_{st}$-index (retrieved from https://omniweb.gsfc.nasa.gov/form/dx1.html) for January 12 – 18, 2022 period. Dates are indicated along the upper abscissa.

---

## Author Comment (AC5)

**RC3**: 'Comment on angeo-2023-27', Anonymous Referee #2, 16 Oct 2023
Dear Anonymous Referee #2,

Thank you very much for your comments.
Your comments and changes in the manuscript are marked in Bright Green.

The presentation of the state of the art and the bibliographic research are both quite good. The English form is almost perfect, even if, probably, the paper would benefit from a reading by a mother tongue English speaker. The topic discussed in the paper is well presented and everything goes smoothly until Section 4. Section 4, Instrumentations and tecnhiques, is, in my opinion the weakest section in the paper. Here the algorithm used for the study is presented, but the level of details is not sufficient. The author should go in much more details, explain how the algorithm was implemented, and provide quantitative statements.

Regarding the technique, it is presented in a mathematical rigorous manner providing a general framework for detecting perturbations from any high-power source of energy. Nevertheless, the technique is based on a clear and simple physical ground: any changes (spikes, most frequently) in the magnetic field strength that arrive necessarily at every point of the observational grid with the same speed, associated with a well-known type of wave, are considered to be caused by the source of energy, if these changes are not observed in the records made under quiet time conditions. The errors are indicated throughout the text. To illustrate the workings of this algorithm, I have already prepared a figure especially for the first referee, Dr. Adrian Hitchman.
     This figure, copied here below, shows UT variations in all 19 *X*-components of the geomagnetic field together, in the distance from the volcano vs UT plane. The vertical dashed line indicates the moment of the volcanic explosion, while the six oblique straight regression lines virtually connect the possible moments of the onset of the magnetic field response indicated by the arrows in Figures 2–20. These variations have already been presented separately in Figures 2–20. Thus, these data clearly show that the disturbance time delay exhibits a tendency to increase with distance from the volcano, which testifies to the disturbance being propagated from the volcano.

[Figure]

Section 5 is, sorry to say, a bit boring. The signals acquired by each of the many stations are presented in Figures, and each of them is commented in the text. This is not the way to proceed. A competent reader can watch the Figure and deduce the most important issues from them. The comment should be collective, and intended to put in evidence the general message, like the ones put in the subsequent sections. I would have expected a different way of treating the data (maybe with figures summarizing the relevant aspects). It is also weird that the author, at some point, start using the word "trend" with a symbol that was never used up to that point (and X with a bar on it, line 197). Anyhow, almost 30 pages are used for this list which are way too much.

Regarding Section 5 being a bit boring, unfortunately, one cannot do without "a bit boring" Section 5. The results of the analysis are described in Section 5. Without this examination, the results of this work would be groundless and unvalidated.

Regarding the comment should be collective, the collective comment is presented in Section 6, Statistical data analysis of the bay excursions in geomagnetic field strengths, and in Section 7, Statistical data analysis of the quasi-periodic variations in geomagnetic field magnitudes.

Regarding the word "trend", Dear Anonymous Referee #2, I thank you for indicating this blunder. The word trend now appear for the first time on page 8 (Line 149) with a symbol with a bar on it.

Last two Sections, where discussion of the results and drawing of the conclusions are done, are much more interesting, and I think that there good results and considerations there. In general, I think that the paper needs a deep revision, getting rid of the central part (to be strongly reduced, and maybe to be moved in an Appendix), but there is something good in it, so my advice is to reconsider it after the revision will be done.

It is hardly advisable to move Section 5 to Appendix. This disrupts the structure of the entire work. Section 5 is the main one, the entire work will not be complete or validated without Section 5. In addition, this idea has not been supported by both Reviewer #1 and a number of community comments.

In the following more specific modifications to be done:

Dear Anonymous Referee #2, I thank you for indicating these modifications, which are marked in Bright Green in the manuscript.

line 42 (Line 46, now): the current density J should not be expressed in ma/m^2 instead of mA/m?

The expression has been corrected.

line 73: "these variations is to advance understanding of this scientific issue"--> "these variations is in order to advance understanding this scientific issue"

line 84: "Table 1. Basic information on volcanos" -->"Table 1. Basic information on largest volcanos eruption recorded"

line 85: the line should not be indented.

line 136: "Figure 1: Map showing the sites of the recording stations."--> "Figure 1: Map showing the sites of the recording stations used for the present study"

line 150: "The more rapid fluctuations"--> "faster fluctuations"

line 443: I guess that R2 is the regression coefficient. Why not calling it like this?

The coefficient $R^2$ is termed the adjusted coefficient of determination. The regression lines are given in Equations (1)–(6).

line 449: "The formation of disturbance is close to root mean square deviations in time delays" what is the meaning of this sentence?

This sentence means that the error in time delay estimates is approximately equal to or greater than the time the disturbance takes to reach ionospheric heights, and consequently, the latter cannot be estimated.

line 463: Here the author discusses results presented in Table 3, shown several pages earlier. I would put this section close to the Table.

Transfer of Section 7 to Table. 3 does not seem appropriate. This is an independent section that is based on digital data from Table 3.

line 522: Please explain where this equation comes from.

With the use of Maxwell's equation

$$\nabla \times \mathbf{B} = \mu_0 \mathbf{j},$$

the perturbation in the magnetic induction, $\Delta\mathbf{B}$, caused by the perturbation in the current density, $\Delta\mathbf{j}$, is given by

$$\nabla \times \Delta\mathbf{B} = \mu_0 \Delta\mathbf{j}.$$

Since the derivatives with respect to the horizontal coordinates are much smaller than with respect to the height, we can write

$$\nabla \times \Delta\mathbf{B} \approx \frac{d}{dz}\Delta B,$$

where $\Delta B$ is the vector components in the horizontal plane. Using scaling arguments, the above equation can be written as

$$\left|\frac{d\Delta B}{dz}\right| \approx \frac{\Delta B}{\Delta z},$$

where $\Delta z$ is the $E$-region dynamo thickness over which the contribution to the magnetic effect is produced.

line 529: "take" --> "takes"

line 530: "occur" --> "occurs"

Citation: https://doi.org/10.5194/angeo-2023-27-RC3

The author is grateful to Anonymous Referee #2 for the valuable comments that have helped the Author greatly improve the draft of his paper.

Sincerely,
Leonid Chernogor.

---

## Author Comment (AC6)

Dear Dalia Burešová,

I have prepared my responses to the reviewer and made appropriate changes to the manuscript marked in green.

Yours sincerely,

Prof. Leonid F. Chernogor
August 17, 2024

Reply to Anonymous Referee #3's comments

Dear Anonymous Referee #3,

Thank you very much for your comments. Author's reply and changes in the manuscript are marked in green.

The author has added some material to the manuscript in this revision, but in my opinion some of the main questions still remain to be addressed before the paper can be accepted for publication. In particular, the analysis presented at the very end of section 7, including figures 24 and 25, is more convincing with regards to identifying the onset time of disturbances. However, this seems to be added in here as an afterthought without much relation to the rest of the manuscript.

Dear Anonymous Referee #3, Thank you very much for this comment. Indeed, you are right. The paragraph discussing the analysis including Figures 24 and 25 lacks a proper topic sentence. We are sorry. Now, this paragraph begins as follows.

Standing sound waves generated near the volcano produce a very important quasi-periodic effect of Tonga volcano termed the acoustic resonance (Chernogor, 2023d). This effect occurs only in the vicinity of the volcano, as well as in the magnetically conjugate region, albeit insignificantly. The magnetic effects of the Tonga volcano explosion observed at other magnetic stations have other physical nature, which is attested by the propagation speeds revealed in this study, and which unambiguously indicate the type of the wave transporting the disturbance.

1) Considering figures 6 & 7: the arrows on these figures are supposed to indicate the onset of disturbances that are possibly the result of the eruption. However, it is still not clear how these arrows are positioned. In the plots, there seems to be nothing special occurring at the marked times.

Dear Anonymous Referee #3, Thank you very much for this comment. Regarding Figures 6 & 7, the algorithm for finding the geomagnetic field response to Tonga volcanic explosion is described in detail in Line 114–130. Line 162 tells that: "on Figures 6 and 7 the arrows mark the possible start and end times of the geomagnetic field response." Naturally, the effect magnitude is relatively small, even for the API station. A particular magnetogram does not permit us to state with confidence that this particular variation is caused by the volcano. Confidence appears as a result of (1) common (systems) analysis of all magnetograms, (2) estimating expected propagation speed of the disturbance, (3) determination of the type of the wave transporting this disturbance. In addition, intercomparisons between the data for 15 January 2022 and the quiet time data exhibits two principle features (easily seen in Figures 6&7). First, the tendency for increasing time delay with increasing distance between the volcano and the magnetometer station. Second, the tendency for increasing the duration of the bay-like disturbances is observed with increasing distance between the volcano and the magnetometer station. Disturbances observed during quiet time period were virtually synchronous.

Regarding the plots, one has to notice that the plots in Figures 6 & 7 are drawn to scale and present the data on bay-like disturbances acquired over a range of distances increasing fifteen-fold, from ~840 km to ~12 000 km. As a result, the weak strength effects are masked against the background of strong strength effects. This is the manifestation of the well-known misleading representation of data (see, e.g., figures 3 & 4 at https://topdrawer.aamt.edu.au/Statistics/Misunderstandings/Misleading-graphs/Misleading-scales). Therefore, only one of the nineteen plots in Figures 6&7 could be correctly represented.

We have arbitrarily changed the scales in Figures 6 for Anonymous Referee #3, and the result is a complete mess:

[Figure]

Nevertheless, the bay-like disturbances became easily visible to the unaided eye, while even the distorted time delay of the effect shows tendency for increasing with increasing distance between the volcano and the magnetometer station, and the distorted duration of the bay-like disturbances exhibits tendency for increasing with increasing distance between the volcano and the magnetometer station.

We have arbitrary changed the scales in Figures 7 in the same way and the result is also a complete mess:

[Figure]

Nevertheless, the bay-like disturbances became easily visible to the unaided eye, while even the distorted time delay of the effect shows tendency for increasing with increasing distance between the volcano and the magnetometer station, and the distorted duration of the bay-like disturbances exhibits tendency for increasing with increasing distance between the volcano and the magnetometer station.

2) Figures 15 & 16: It is not evident that the disturbances circled on these figures are more significant than other variations at the same observatory. For instances: for the EYR station in Figure 15 the variation marked after 10 UT are very small compared to those at other times; similar for KDU in Figure 16 around 12 UT; and so on.

Dear Anonymous Referee #3, Thank you very much for this comment. Figure 15 clearly shows that the amplitude and period during period 10:10–11:05 UTC are sharply different from those observed at other moments of time. Regarding the KDU station, the amplitude and period observed during the interval ~10:00–~13:00 UTC are significantly different from the amplitude and period observed during ~13:00–14:45 UTC. It is the last oscillation that is most likely caused by the volcano.

While analyzing temporal variations in the strength of the geomagnetic field, one ought to remember that many energy sources affect the geomagnetic field. Therefore, the systems approach, which has been employed in this study, is a sole approach permitting the disturbances accompanying a particular source to be determined.

3) It is also not entirely clear how the strengths of fluctuations shown in figures 15 & 16 were obtained. Was this done by removing a running average, subtracting a long term median, ...?

Dear Anonymous Referee #3, Thank you very much for this comment. To reveal the rapid fluctuations and oscillations, the slow trend, determined as moving average, is subtracted from the raw data (see Line 135–136: To

discern rapid fluctuations and oscillations, the 120-min moving average (trend) was calculated in a 1-min step first. Further, this trend was subtracted from the raw data temporal dependences.).

4) Because it is still not clear how the onsets of various disturbances at the different stations are determined, it is also not clear how the statistics in sections 6 and 7 were obtained. However, this is not an independent issue: if the previous points are resolved the statistics should become clear as well.

Dear Anonymous Referee #3, Thank you very much for this comment. Thus, Points 1 and 2 explain how the statistics is obtained.

The author is grateful to Anonymous Referee #3 for the valuable comments that have helped Author greatly improve the draft of his paper.

Sincerely,
Leonid Chernogor.